# Joining of metallic glasses in liquid via ultrasonic vibrations

Luyao Li[1], Xin Li[2], Zhiyuan Huang[3,4], Jinbiao Huang[1], Zehang Liu[1], Jianan Fu[5], Wenxin Wen[1], Yu Zhang[1], Shike Huang[1], Shuai Ren[1] & Jiang Ma ®[1] ✉

Joining processes especially for metallic materials play critical roles in manufacturing industries and structural applications, therefore they are essential to human life. As a more complex technique, under-liquid joining has far-reaching implications for national defense, offshore mining. Furthermore, up-to-now, the effective joining of metals in extreme environments, such as the flammable organo-solvent or the arctic liquid nitrogen, is still uninvestigated. Therefore, an efficient under-liquid joining approach is urgently called for. Here we report a method to join different types of metallic glasses under water, seawater, alcohol and liquid-nitrogen. The dynamic heterogeneity and liquid-like region expansion induces fluid-like behavior under ultrasonic vibration to promote oxide layer dispersion and metal bonding, allowing metallic glasses to be successfully joined in heat-free conditions, while still exhibiting excellent tensile strength (1522 MPa), bending strength (2930 MPa) and improved corrosion properties. Our results provide a promising strategy for manufacturing under offshore, polar, oil-gas and space environments.

Joining processes play a significant role in almost all manufacturing industries and for structural applications[1,2], therefore, they are essential to various aspects of human life. A large number of techniques are available for joining in atmosphere[2–4], however, many of them cannot be applied in liquids such as offshore and marine applications, flammable and explosive environments or extreme low-temperature conditions. In these environments, researchers have studied underwater welding joining technology most commonly, including underwater welding, which is the most representative[5–7]. Generally, underwater welding techniques can be classified as wet welding[8] and dry welding[7]. Compared with the dry welding, which requires large equipment support and high involved costs[9,10], simplicity of the wet welding makes this process possible to weld even the most geometrically complex structures[5,8]. Therefore, the wet welding underwater has been considered as the most effective, efficient and economical method to repair ocean platforms and is essential for the offshore industry in ship construction, oil & gas mining in the middle of the sea and nuclear

power plants[6,8,11]. Conventional underwater welding techniques such as shielded metal arc welding, tungsten inert gas welding and metal inert gas welding[12,13] were developed and advanced techniques such as friction welding[14,15] and laser welding[16,17] were also invented to improve and perfect this promising process.

However, the grand challenges facing us are arduous in the under-liquid joining. Let us take underwater wet welding for example, the high working temperature of 500 to 800 °C and the resulting ultrahigh cooling rate would cause the deterioration of mechanical performance of welded region[6,18–20]. Importantly, the pores were inevitably formed by molecular hydrogen, carbon monoxide or water vapor under such harsh working environments[21,22]. In addition, the large electric current (-hundreds of ampere)[23] needed to melt the bare metal could bring serious life risk of electric shock to the welder/diver[24]. To even worse, the stability of the arc is also affected by increasing water depth[20,22]. Most notably, the now available joining techniques were only suitable for the underwater environment, when joining works are needed in

[1]Shenzhen Key Laboratory of High Performance Nontraditional Manufacturing, College of Mechatronics and Control Engineering, Shenzhen University, Shenzhen 518060, China. [2]School of Mechanical, Electrical and Information Engineering, Shandong University, Weihai 264209, China. [3]Songshan Lake Materials Laboratory, Dongguan 523808, China. [4]School of Materials Science and Engineering, Shanghai University, Shanghai 200444, China. [5]Department of Mechanics and Aerospace Engineering, Southern University of Science and Technology, Shenzhen 518055, China. ✉e-mail: majiang@szu.edu.cn

flammable liquid conditions (such as the repair in the oil, gas and organic solvent storage containers), they could totally fail. Furthermore, joining in cryogenic environments, such as liquid nitrogen (−196 °C), was challenging. It is well known such extreme temperature was critical trouble for space working (spacecraft repair in Earth orbit, the Moon and Mars)[25]. Although a few studies have reported that friction welding[26,27] and laser welding[28] were used in liquid nitrogen, this method either only uses liquid nitrogen as a means of quick cooling[26] or the equipment is not as portable enough[26,28]. Therefore, to address all these challenges, the invention of effective wet joining techniques is of huge significance for the increasing and stricter demand from various under-liquid joining processes in the fields of national defense, offshore mining, energy storage, space exploration et al.

As a promising material for applications, metallic glasses (MGs) exhibit high strength[29], high elasticity[30], superior wear resistance[31], and high corrosion resistance[32]. Therefore, joining MGs in liquid environments will have significant applications. At present, there are two main mechanisms for joining MGs. One involves melting MGs and subsequent rapid quenching, mainly including laser welding[33], explosion welding[34] and electric pulse welding[35]. Another potential method is joining in the supercooled liquid region, such as thermoplastic joining[36] and friction stir welding[37]. These methods also require high heat input and complex devices, making them difficult to apply in harsh liquid environments. Therefore, up to now, there is no one technology that can truly implement MG/MG and MG/non-MG joining under liquid. In recent years, a promising joining method has emerged - ultrasonic vibration joining (UVJ) technology[38,39]. Generally, ultrasonic-related technology was used in the field of welding for plastics[40,41] and low melting metals[41], except for these, more often as an auxiliary technology to other welding methods[42–44], which is also difficult to be applied in various liquid environments due to the temperature-dependent mechanism. However, due to the characteristics of low-temperature requirements, fast joining process, and simple operation when joining MGs[39], UVJ technology has great potential for realizing joining MGs in liquid environments.

Here, we report an under-liquid joining strategy that involves the use of UVJ technology. In this work, the ultrasonic vibrations (20000 Hz) are utilized to join MGs and MG/non-MG joining parts under pure water, seawater, alcohol and liquid nitrogen environments conveniently. Compared to common under-water joining technology[6,18–20], our method has the advantage of no temperature rise, which will avoid the effects of excessive temperatures as well as cooling rates on the joined properties. Moreover, as a process that no electric arc is involved, the instability of the power supply system and the safety of electricity use are no longer a concern. In the joining process, the dynamic heterogeneity and liquid-like region expansion induces fluid-like behavior under ultrasonic vibration to promote oxide layer breakage and metal bonding, allowing MGs to be successfully joined without obvious seams. After UVJ, the joined samples not only still maintain 94% of the tensile strength (1522 MPa) and bending strength (2930 MPa) of the as-cast sample, but also exhibit improved corrosion properties. This work not only opens a window for underwater joining technologies, but also provides a feasible strategy for extreme environments like flammable environments in oil, gas, organic solvents and cryogenic conditions in space.

## Results and discussion
### Joining under liquid environment
The schematic diagram of under-liquid UVJ is shown in Fig. 1a, in which the ultrasonic equipment consists of a control unit (adjusting vibration parameters) and a sonotrode (releasing ultrasonic vibrations). The joining process is relatively simple. We first place the MGs into the clamping device to hold them and then apply a low static pre-pressure (200 N) by ultrasonic sonotrode. Afterward, the sonotrode releases

vibrations at a frequency of 20,000 Hz. The shape of the vibration in sonotrode presents a sinusoidal function with an amplitude of 44.4 μm. The displacement of the ultrasonic sonotrode with respect to time $t$ can be expressed by the equation $d(t) = A\sin(2\pi f \cdot t)$, where $A$ and $f$ denote the amplitude and frequency of the ultrasonic vibration, respectively.

To verify the applicability of UVJ in liquid environments, 4 different liquid environments (pure water, seawater, alcohol and liquid nitrogen) and 4 usual joining types (butt joint (cylindrical), butt joint (sheet), lap joint and T-joint) were chosen. The display diagrams of the Zr-based joined samples are shown in Fig. 1b−e. It can be observed that all samples are tightly joined and spillage existed at the join interface, the spillage was caused by the softening of the MGs[45]. Meanwhile, UVJ in liquid environments can also be applied to MGs of other compositions of MGs (such as $La_{55}Al_{25}Ni_5Cu_{10}Co_5$) (see Supplementary Fig. 1). In addition, the present work realizes the joining between $Zr_{55}Cu_{30}Al_{10}Ni_5$ MG and TiZrHfBeNi high entropy-bulk metallic glass (HE-BMG) (as shown in Supplementary Fig. 1), indicating that UVJ technology is not limited to only the joining of the same MG material.

To research the joining process in detail, the high-speed camera was used to capture the slow play process of Zr-based (Supplementary Movies 1-3) and La-based (Supplementary Movies 4-6) samples joining under various environments. Due to the details of the MGs was difficult to be photographed under the liquid nitrogen environment by using the high-speed camera, so the operation steps of joining in liquid nitrogen were captured by the normal camera (Supplementary Movie 7). Figure 2a displays partial pictures of the joining process (joining Zr-based MGs in seawater as an example). It is noteworthy that the surfaces of the two bulk metallic glasses (BMGs) spit out high amounts of bubbles under the ultrasonic vibration, demonstrating the violent vibration between the two surfaces. After that, a huge amount of flowing deformed MGs overflow at the interface and then two MGs realize a tight connection. The thermocouples were used to measure the real-time temperature situation of the joined surface. The temperature curves show that joining Zr-based MGs in 4 environments exhibited similar instantaneous maximum temperatures: 89.0 °C in pure water, 110.1 °C in seawater, 92.6 °C in ethanol and 68.8 °C in liquid nitrogen, respectively (Fig. 2b). The overall temperature distribution of join in liquid nitrogen exhibits lower temperature than the other 3 environments, mostly below 0 °C. The above results show that the in-processing temperatures are far below the glass transition temperature (about 390 °C for $Zr_{55}Cu_{30}Al_{10}Ni_5$ MG), demonstrating that the joining of MGs under ultrasonic vibrations was a cold process. In addition, the process of joining La-based MG in liquid environments also exhibited similar temperature rules in Supplementary Fig. 2. The stress-time curves of the UVJ process measured by the force gauge indicate that the joining process in all environments maintains low loads (around 20 MPa) (Fig. 2c).

In order to initially detect the joining quality, the X-ray diffraction (XRD) and the computed tomography (CT) analysis were used to determine the amorphous properties and bonding situation of the joined samples, respectively. The XRD patterns show that the Zr-based MG remains amorphous state after joining, which means that the material can still maintain the excellent properties of the special amorphous structure (Fig. 2d). Figure 2e shows the six longitudinal-sections CT images of homogeneous as well as heterogeneous joined samples at different cutting positions, and it can be clearly seen that the joined seam almost completely disappears, indicating the effective bonding of the MGs. In order to more clearly express the combination of two BMGs, the CT diagram of the heterogeneous joined sample was represented in the form of relative density distribution diagrams (as shown in Fig. 2f), where the red part represents $Zr_{55}Cu_{30}Al_{10}Ni_5$ MGs and the green represents TiZrHfBeNi HE-BMGs. Notably, it can be clearly seen that the two MG phases mutually diffuse at the interface and form a tight metallurgical bond. The

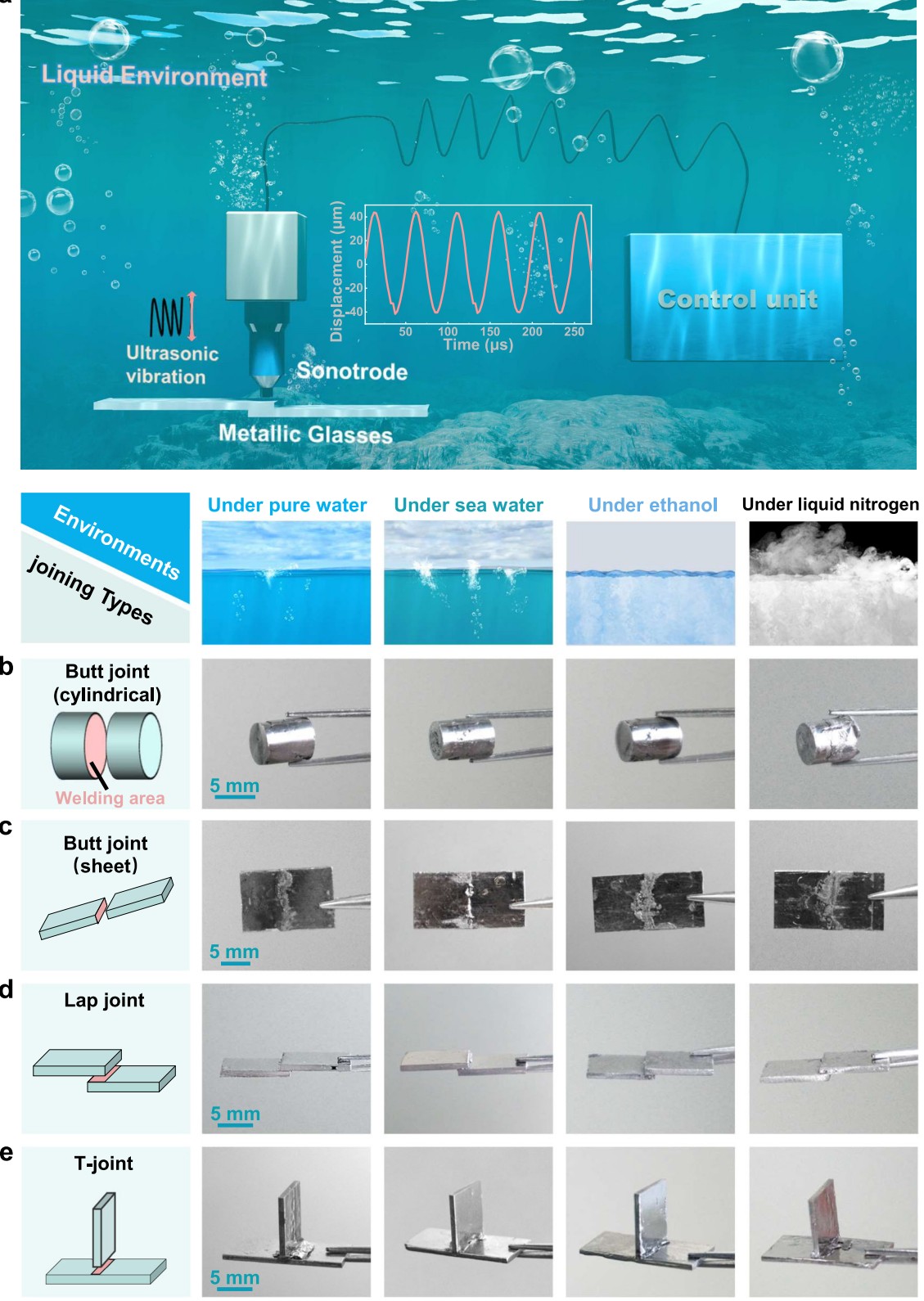

**Fig. 1 | The display of multiple joining types of MGs. a** The schematic diagram of joining MGs under liquid environments. **b, c** Samples display diagram of 4 different types of joining types: **b** butt joint of cylindrical, **c** butt joint of sheet, **d** lap joint, **e** T-joint.

heterogeneous joined sample scanned from the cross-sectional direction also shows a similar situation, as shown in Fig. 2g. Meanwhile, the joining in liquid environment was compared with air, as shown in the Supplementary Fig. 3. Due to the dissipation of ultrasonic energy, joining in liquid environments requires more energy than in air. The above results showed that under appropriate welding parameters, all environments (including air) can ultimately achieve completely defect-free joining.

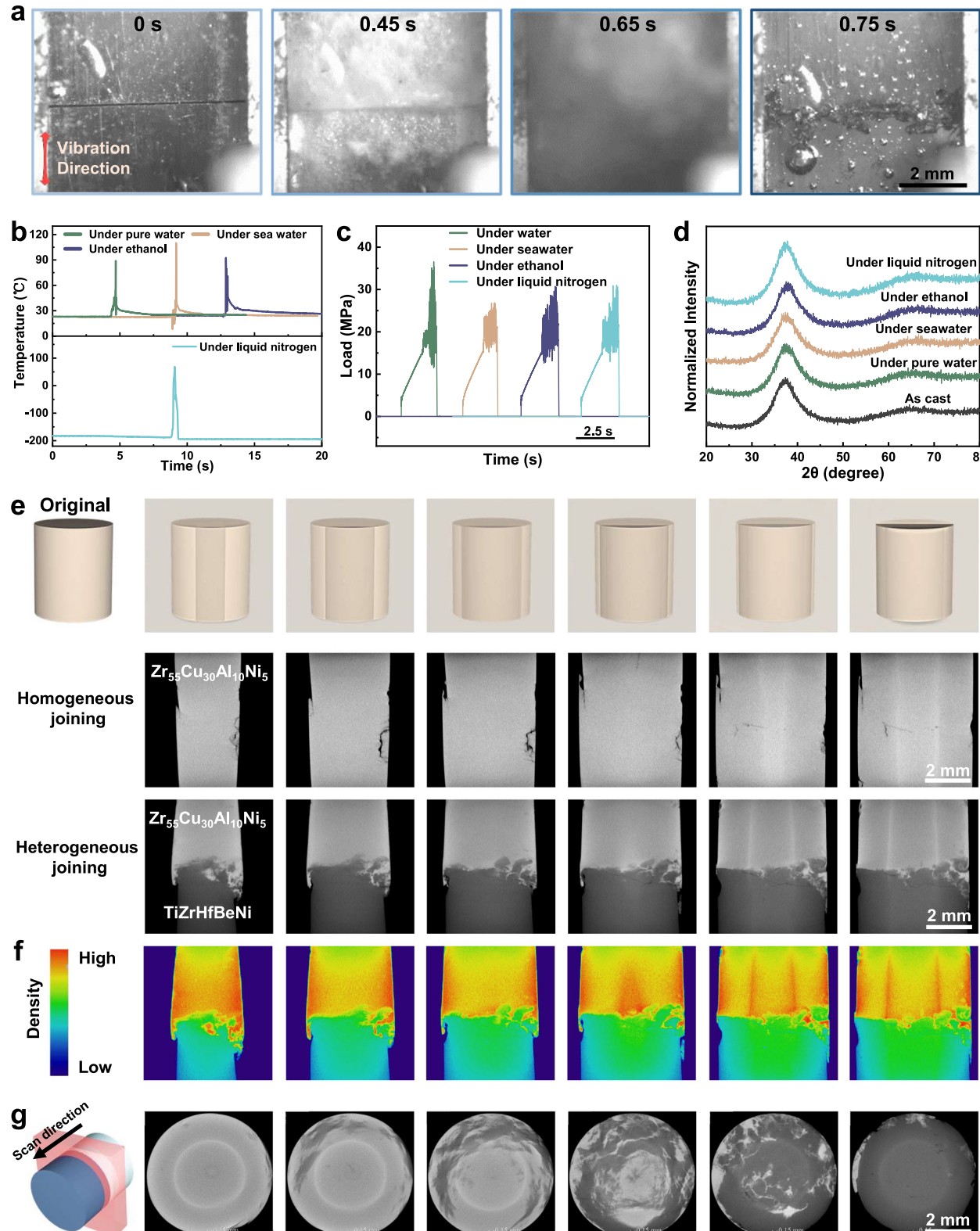

**Fig. 2 | The characterizations of joining processes and effects. a** The joining process in liquid environments captured by the high-speed camera (with seawater as an example). **b** The temperature-time curve of the Zr-based join measured by the thermocouple. **c** The stress-time curve of joining in 4 types of liquid environments detected by force gauge. **d** The X-ray diffraction (XRD) patterns of as cast MGs and 4 liquid environments joined samples. **e** The computed tomography (CT) scan patterns of homogeneous and heterogeneous joined samples. The inset shows the scanning direction (longitudinal-section). **f** The density distribution diagram of heterogeneous joined sample corresponding to e. **g** The CT scan patterns of heterogeneous joined samples. The inset shows the scanning direction (cross-section).

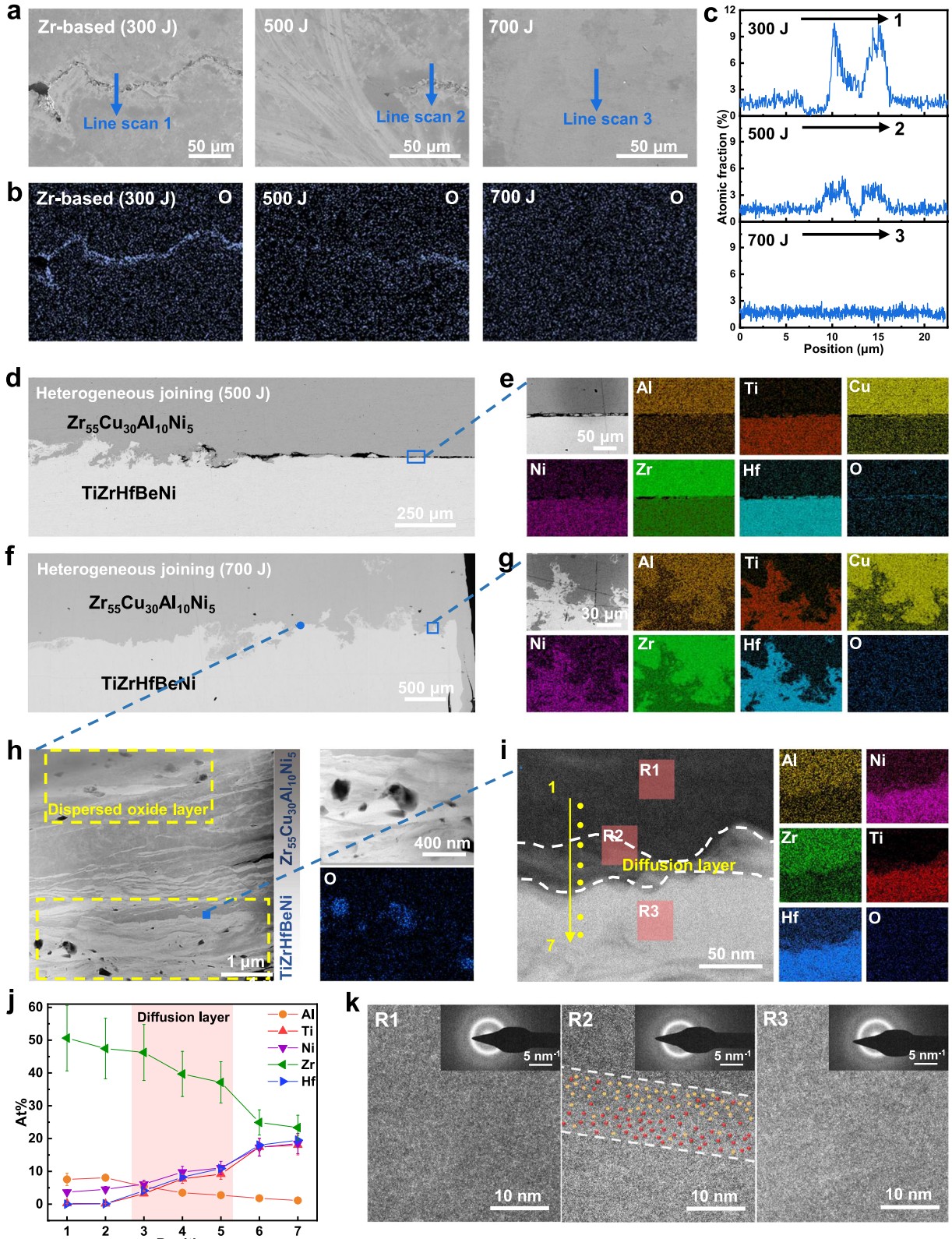

## Joining mechanism

In order to investigate the mechanism of joining in liquid environments, Fig. 3 analyzes the micro-morphology evolution at the interface of the joined sample. The cross-sectional field emission scanning electron microscope (SEM) micrographs in Fig. 3a show the Zr-based MG joined interfaces under different ultrasonic energy (butt joint of cylinders as an example). It can be clearly seen that as the energy of

ultrasonic vibration increases from 300 J to 700 J, the joined seam gradually decreases and even disappears. In the sample of 700 J, there is without any trace of voids and cracks, denoting the formation of metallurgical bonding. The energy-dispersive X-ray spectroscopy (EDS) analysis was used to research element distribution of that joining process. It is noteworthy that oxygen elements are abundant in the voids and cracks in the distribution map of 300 J sample, while the

**Fig. 3 | The microscopic characterization of joined interfaces. a, b** The field emission scanning electron microscope (SEM) morphology and corresponding oxygen element distribution images of the Zr-based homogeneous joined sample after 300 J, 500 J and 700 J ultrasonic vibration joining (UVJ) treatment. **c** The energy-dispersive X-ray spectroscopy (EDS) line scan image of the oxygen element content through the joined seam. The scanning direction is shown in a. **d, e** The SEM morphology and corresponding EDS mapping images of the heterogeneous joined sample after 300 J UVJ treatment. **f, g** The SEM morphology and corresponding EDS mapping images of the heterogeneous joined sample after 700 J UVJ treatment. **h** The high angle annular dark field (HAADF) at the interface of the

heterogeneous joined sample. The inset shows the EDS mapping of oxygen elements in part of regions. **i** The HAADF and corresponding EDS mapping at the diffusion layer of heterogeneous join. **j** The nanobeam energy-dispersive X-ray spectroscopy (EDX) through the joined interface. The scanning point is shown in i. The error bars represent the systematic relative errors of elemental content obtained from the nanobeam EDX measurement. **k** High-resolution image and selected-area electron diffraction (SAED) patterns corresponding to the R1, R2, and R3 regions in i. The red and yellow spheres in k schematically represent the diffusion of atoms at the diffusion layer.

oxygen elements of the 700 J specimen are uniformly distributed (Fig. 3b). The EDS line scan curves across the joined interface also show that the elemental oxygen content at the interface gradually decreases with the increasing energy, as shown in Fig. 3c. The above finding indicates that the joining process under UVJ can effectively break through the oxidation layer to form the firm metallic bonding.

To express more clearly the metallurgical bonding of MGs, the underwater heterogeneous joining between $Zr_{55}Cu_{30}Al_{10}Ni_5$ MG and TiZrHfBeNi HE-BMG was researched. Figure 3d shows cross-sectional SEM micrographs of the 500 J sample, where some parts are bonded but a lot of regions still exist seam. The EDS analysis of the seam region (Fig. 3e) shows that there is no significant diffusion between local poorly joined interfaces. Similar to the homogeneous joining, an increase in energy will lead to the disappearance of the oxygen-rich seam (Fig. 3f and g), resulting in a tight bond. However, it is noteworthy that not only did the interface of the 700 J sample exhibit an irregular interface morphology, but also a mixture of the two MGs was observed at a more microscopic scale. The EDS analysis of the microscopic region in the 700 J interface (Fig. 3f) also proves the mixing of the two MGs. In other environments, the above phenomenon was also found, as shown in Supplementary Fig. 3. The above results clearly demonstrate that UVJ is a process of mixing the fresh interface after the "disappearance" of the oxide layer. Now, the question is, where did the oxide layer go, and how did the two MGs bond?

To explore these questions, the transmission electron microscope (TEM) was used to observe the joined interface at a finer scale (Fig. 3h-k). The results show that not only the interface has achieved perfect metallurgical bonding, but also an intense nanoscale mixing zone existed between the two MGs (Fig. 3h). And more importantly, a large number of dispersed oxide particles were observed around the interface, which proves the crush of the oxide layer under UVJ. Except for a few large particles, most of these particles have been crushed to a considerably smaller size, more analysis of oxide particles is shown in the supplementary Fig. 4. A selected interface was observed in Fig. 3i, where the mutual diffusion of elements at the two-phase interface can be clearly seen, and the diffusion layer can reach up to about 50 nm. Meanwhile, nanobeam EDX with a scanning diameter of 2 nm was used to analyze multiple points at the interface (Fig. 3j), in which the quantitative element analysis at the interface proves the existence of the diffusion layer. In addition, the high-resolution and diffractive halo-ring patterns (Fig. 3k) indicate completely amorphous features in both the interface region and the respective bodily regions of $Zr_{55}Cu_{30}Al_{10}Ni_5$ MGs or TiZrHfBeNi HEA-BMGs. The red and yellow spheres in Fig. 3k schematically represent the diffusion of atoms at the diffusion layer. Based on the above observations, it can be concluded that the strong mixing of the two phases and the dispersion of the oxide layer are related to the phenomenon of fluid-like behavior occurring in the joining interface (see Supplementary Movies 1-6).

What causes the fluid-like behavior of MGs? The literature shows that the MGs consist of liquid-like regions and solid-like regions, which is referred to as the structural heterogeneity of BMGs[46]. The solid-like region forms elastic networks, while the liquid-like region enveloped by the network acts as a viscous flow unit that dissipates energy. In

BMGs, local shear events induced by external forces have been shown to cause expansion of the liquid-like region after absorbing energy, which promotes the growth and formation of the liquid-like region[47,48]. Now the question is: what will happen when under cyclic loading? The literature[49] has investigated cyclic loading-induced liquefaction occurring in saturated soils with similar structural heterogeneity, providing us with insights. The principle is that cyclic loading causes high pressure concentrated around the liquid-like regions, which continuously accumulates due to no sufficient time for stress relaxation. When MGs are subjected to cyclic loading, this activation and expansion of the liquid-like regions also will become more obvious[45]. The above effect leads to extensive expansion of the liquid-like regions in a short time. When a sufficient number of fluid-like regions has accumulated and linked together, the connecting elastic network will be broken, resulting in viscous behavior and softening of the BMGs (Fig. 4a).

To validate the proposed MG softening mechanism, we used dynamic scanning probe microscopy (DSPM) to mapping the viscoelastic loss tangent (tan$\delta$) of $Zr_{55}Cu_{30}Al_{10}Ni_5$ MG, where $\delta$ represents the phase shift between the dynamic force and amplitude. The tan$\delta$ is commonly used to correspond to internal friction[50], similar to the results obtained from conventional dynamic mechanical analysis (DMA). Figure 4b and c show the tan$\delta$ distribution images obtained at the driving frequency of 200 Hz and 72,000 Hz, respectively. The dynamic heterogeneity of the MGs can be observed from those figures. On the other hand, as the frequency increases from 200 Hz to 72,000 Hz, the average tan$\delta$ increases sharply from 0.130 to 0.229 (Fig. 4d), which indicates that the MG atoms have absorbed a large amount of mechanical vibration energy and have been highly activated. Such a high internal friction (tan$\delta$ = 0.229) is already close to the internal friction exhibited by BMGs after crossing the glass transition point[30], so quick atomic movement and atomic diffusion will be existed[51,52]. Therefore, in this work, fluid-like MGs are easy to interdiffuse, which will lead to the formation of tight bonds. To more intuitively observe the influence of high-frequency loading on MGs softening, a simple Maxwell model can be assumed here[38]. In this viscoelastic model, the relationship between relaxation time ($\tau$) and tan$\delta$ can be obtained[53]: $\tau = 1/(\omega \tan\delta)$, where $\omega$ represents the angular frequency of the sinusoidal variation. Meanwhile, in such a model, relaxation time and viscosity have been proven to be proportional[54], so comparative data of viscosity (or relaxation time) at different frequencies can be obtained (as shown in Fig. 4e). In Fig. 4e, the viscosity (or relaxation time) peak position at $f$ = 200 Hz was normalized. The results show that as the frequency increases from 200 Hz to 72,000 Hz, the viscosity (or relaxation time) of the MGs decreases by three orders of magnitude.

Under enough energy, after the MGs near the interface are completely softened, high-intensity ultrasonic vibration applied to liquids can generate cavitation and acoustic streaming effects[42,55]. After the collapse of these cavitation bubbles, high-intensity shock waves and strong convection will be generated in the fluid. Acoustic streaming was one type of turbulence generated near the solid-liquid interface. These effects promote the broken and dispersion of the brittle oxide layer. On the other hand, the ultrasonic-induced flow provides strong

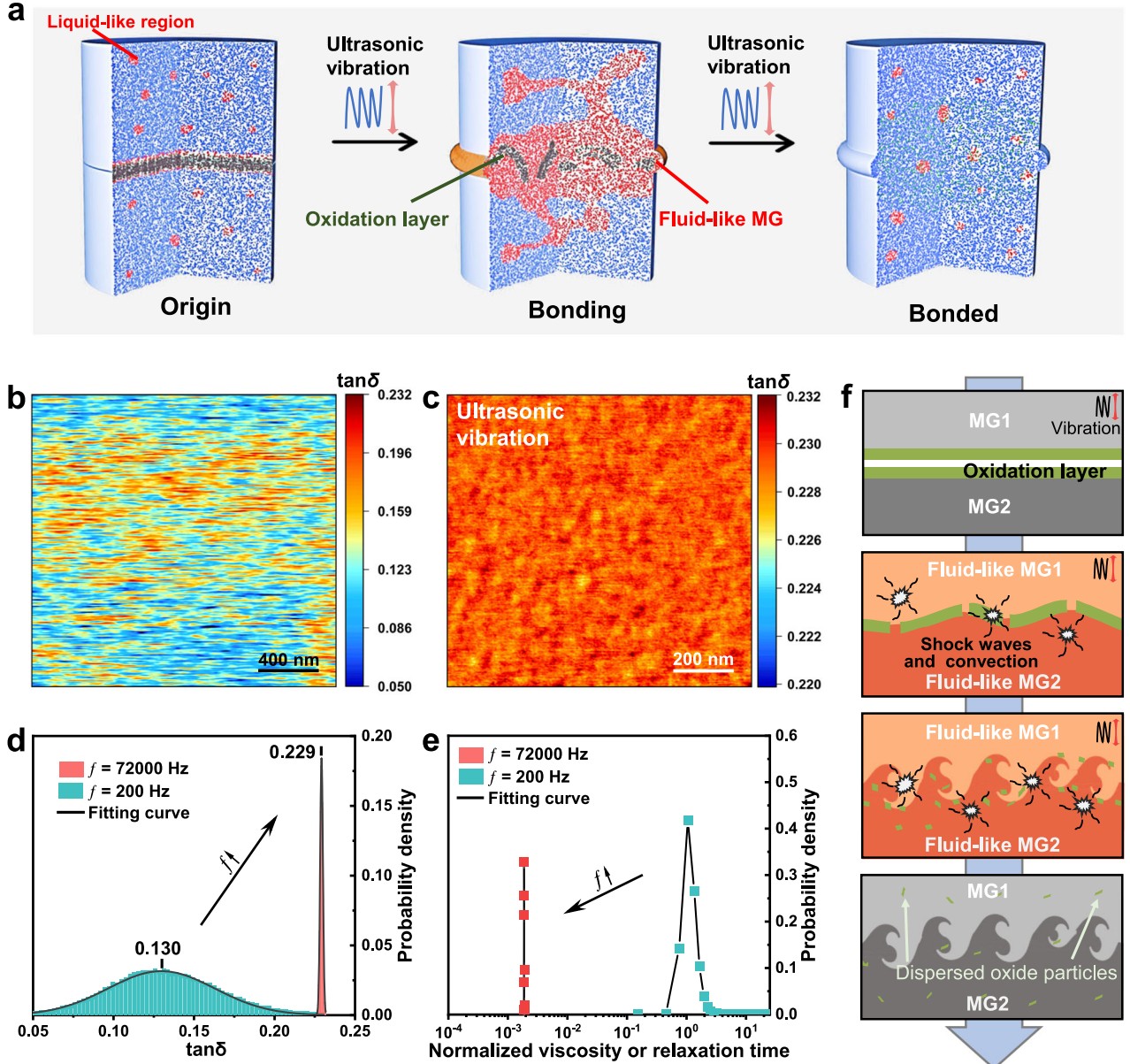

**Fig. 4 | Joining mechanisms. a** Schematic diagram of ultrasonic vibration-induced softening and joining of the interface of MGs (atomic scale). **b**, **c** The viscoelastic loss tangent map at $f$ = 200 and 72,000 Hz. **d** The statistical analysis of b and c with a good fit of Gaussian distribution. **e** Viscosity (or relaxation time) distribution after normalizing the value of the peak position at $f$ = 200 Hz. **f** Schematic diagram of the mechanism for ultrasonic vibration joining (UVJ) with irregular interface morphology.

mixing of two fresh high-energy state fluid MGs, which promotes full contact of the interface and generates an irregular interface. The above process promotes the formation of a good joining interface, as shown in the schematic Fig. 4f. After the joining process is completed, the migration rate of atoms in the liquid-like region decreases due to the removal of the ultrasonic excitation source, resulting in the fracture of the connections in the liquid-like region and the formation of new solid-like regions (Fig. 4a).

**Performance after joining**

To further demonstrate the reliability of under-liquid joining technology, various performance tests were performed on the joined samples, as shown in Fig. 5. Figure 5a shows the Vickers hardness distribution in the section of the Zr-based joined sample. In this case, 20 points on a line perpendicular to the joined seam were selected for the Vickers hardness test. One can see that the hardness has a slight increase with the increase of joining energy in the joined seam, and 700 J samples can reach the same hardness with the as cast MG. To investigate more microscopic properties, the hardness and modulus of the joined interface and the as-cast Zr-based MG were measured by nanoindentation experiments (Fig. 5b, c). At the equivalent pressure, the indentation depth at the joined interface exhibits not much difference than that of the as-cast MG (Fig. 5b). Meanwhile, the specimens also show similar hardness and modulus, as shown in Fig. 5c.

Among all mechanical properties, tensile and bending properties are the most crucial. The experimental results show that the strength of the as-cast sample is 1615 MPa. However, the joined sample can reach an astonishing 1522 MPa, which is 94.2% of the as cast sample (Fig. 5d). At the same time, SEM was used to observe the break surface morphology of the tensile parts, and it was found that the joined sample show close morphology with the as cast MG (Fig. 5e). Actually, small amounts of poor joined interfaces could be

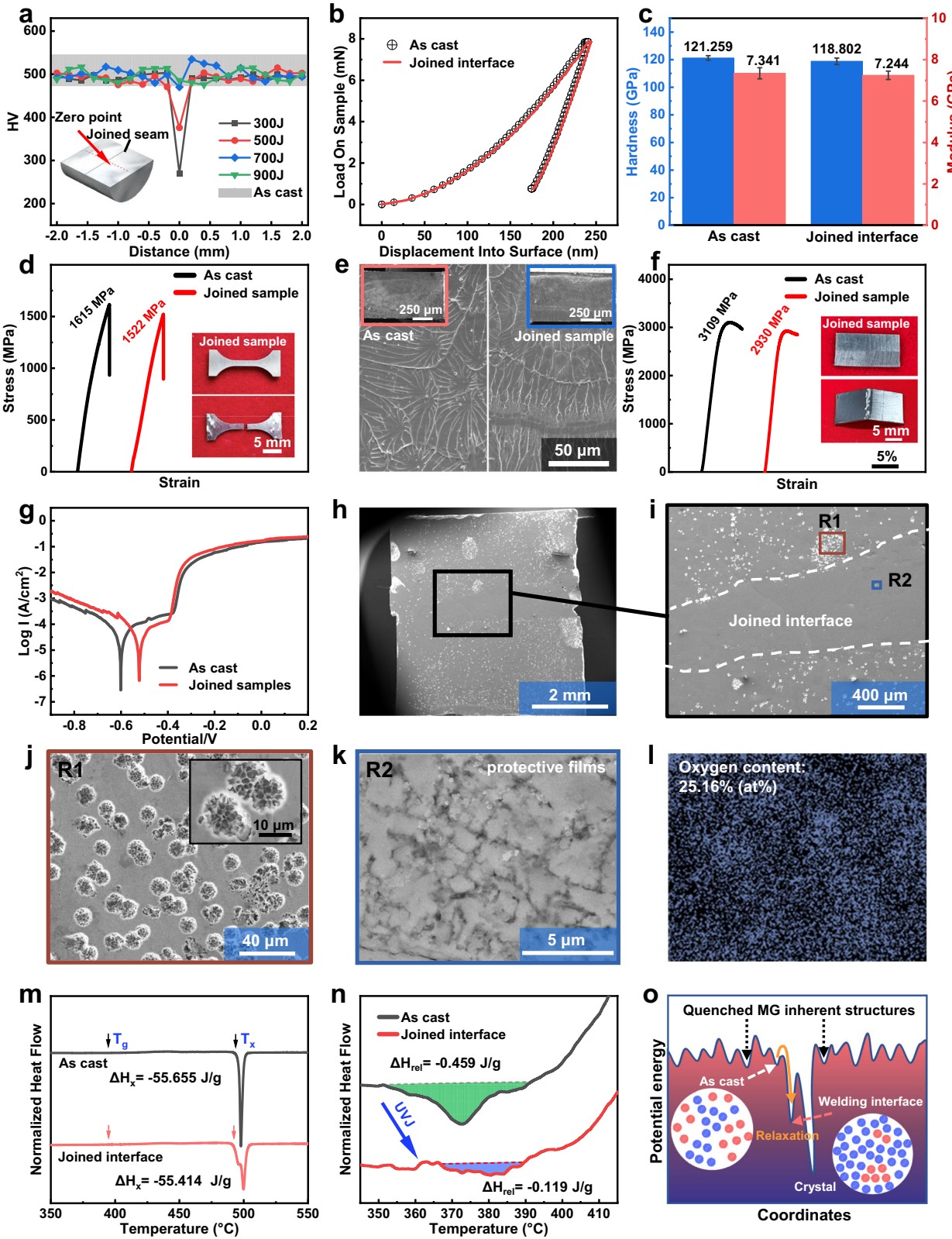

found in break surface morphology, as shown in Supplementary Fig. 5, which demonstrated the fracture surface still remained near the interface. If the welding energy is reduced, the tensile strength will decrease and the poorly joined interfaces will become more pronounced. Meanwhile, three-point bending tests were conducted, with the joined samples exhibiting strength (2930 MPa) comparable to those of the as cast samples (3109 MPa) (Fig. 5f). The above results

prove that the joining samples possess excellent mechanical properties.

In complex liquid environments, the joined materials often require more excellent corrosion resistance than in air. To test the corrosion resistance of MGs after joining, electrochemical polarization experiments were conducted under 1 M HCl environment. The experimental results showed that the joined MG of Zr-based showed a

**Fig. 5 | Performance of joined MGs. a** The micro-hardness test results of 20 dots in the longitudinal section of Zr-based joined samples under different energy. **b** Load-displacement curve for as cast Zr-based MG and joined interface. **c** Comparison of the hardness and modulus between the as cast Zr-based MG and joined interface. The error bars in c represent the standard deviations of the measured values (n = 10). **d**, **e** Stress-strain curves and corresponding break surface morphology of the as cast Zr-based MG and joined sample under room temperature tensile. **f** Bending strength comparison for as cast Zr-based MG and joined sample. **g** Electrochemical polarization curves of as cast Zr-based MG and joined sample in 1 M HCl solution. **h**, **i** the SEM images of the corroded surface of the joined sample.

**j** Large magnification of pits in the non-interface areas (corresponding to R1 in i), which shows a 'honeycomb' structure. **k**, **l** Large magnification of protective films and the corresponding energy-dispersive X-ray spectroscopy (EDS) oxygen content analysis in the interface areas (corresponding to R2 in i), which demonstrates that a protective film has formed. **m**, **n** The differential scanning calorimetry (DSC) traces and the close-up view of the relaxation exotherm. The vertical axis is in arbitrary units. **o** Schematic representation of the structural evolution and stabilization processes under ultrasonic vibration joining (UVJ) in a potential energy landscape. The as cast MG is quenched at a relatively high energy. After UVJ, it can cross an energy barrier and stabilize to a higher-density atomic stacking state.

higher corrosion potential (0.1 V) than the as cast MG (Fig. 5g), showing difference with the common metal that exhibits worse corrosion performance after joining. To research corrosion in detail, the specimens were anodized until pitting corrosion occurred for a period of time and were immediately removed for morphological examination under SEM. It is worth noting that the joined interface of the joined samples did not show significant corrosion (Fig. 5h, i), which is clearly different from the intense pitting corrosion presented by the non-interface region. A more microscopic observation of the corrosion spots in the non-interface region revealed a honeycomb-like structure, indicating intense corrosion (Fig. 5j). However, it was found that only a few areas were corroded at the interface, in which showed significant protective films (Fig. 5k) with high oxygen content (Fig. 5l). At the same time, the corrosion performance of heterogeneous joined samples was also analyzed, as shown in Supplementary Fig. 6. The results show that TiZrHfBeNi has much better corrosion resistance than $Zr_{55}Cu_{30}Al_{10}Ni_5$, and we also found that the portions closer to the interface will exhibit better corrosion resistance, particularly in the two-phase mixing zone. The above results show that UVJ not only did not decrease the corrosion resistance of the joined MGs, but even improved the corrosion performance of the joined interface substantially.

In order to explore the mechanisms of improved corrosion performance, the EDS under SEM was used to measure the elemental distribution of the as cast sample and the joined sample cross-sections. The evidence showed that the elemental content not only had no segregation in the interface and non-interface area in the joined sample, but also was almost consistent with that of the as-cast samples (Supplementary Fig. 7). Obviously, the improvement in corrosion performance must not be caused by the change in composition, and it is more likely induced by structural factor. To investigate this issue, the DSC analysis was performed on the joined interface and the as cast material. The DSC results of the as-cast sample and joined interface exhibited that the various parameters (glass transition temperature ($T_g$), crystallization temperature ($T_x$), crystallization enthalpy ($\Delta H_x$) did not exist significant difference (Fig. 5m). However, the relaxation enthalpy ($\Delta H_{rel}$) shows a significant decrease (from 0.459 J/g to 0.119 J/g), as shown in Fig. 5n.

Numerous studies have proven that the optimization of the pitting behavior of MGs is closely related to its amorphous structure state[56]. In the past years, a large number of studies to optimize the corrosion resistance through the relaxation of MGs have been reported[56–58]. That is due to the fact that the relaxation phenomenon reduces the average atomic spacing and chemical potential of MGs, allowing them to reach a more stable state[56,57], which will greatly improve corrosion resistance. In this work, the joined interface exhibited quite significant relaxation compared to the as cast MG (Fig. 5n). Under ultrasonic vibration, liquid-like regions can be rapidly activated (especially at the interface of the two MGs), leading to an acceleration of the relaxation process and a more intensive arrangement of these regions[59]. As presented in the schematic of energy landscapes (Fig. 5o), the activated configurations jump out the saddle point into neighboring basins with a lower potential energy. The above processes allow the MGs to reach a more stable state, which will benefit their corrosion resistance[56,57].

## More complex joining types

Generally, joining techniques in liquid and even space environments require fabricating more complex joining structures. In order to verify the feasibility of our presented technical approach, we conducted a further exploration. Except for the joining types shown in Fig. 1, UVJ can also join complex structures (stacked structure and frame structure for example) under liquid environments (Fig. 6a, b).

In various environments, except for MG with MG joints, MG with other non-MG parts and non-MG parts with non-MG part joints of a wider range of applications. We already know that ultrasonic vibrations lead to structural changes and the eventual collapse of solid-like regions within the MGs, hence exhibiting liquid-like flow deformation behavior[45]. The deformed flowing MGs can be embedded in parts such as threads to form tight joints (Fig. 6c). Even when observed under SEM, the thread-to-MG joint fails to see significant voids (Fig. 6c), which proves that this joining type is quite reliable. Furthermore, MGs can be used to achieve the riveting of steel parts by deformation of MGs under ultrasonic vibration. This MG-based rivet will maintain the excellent properties that are inherent in MG materials (Fig. 6d). The above evidence demonstrates that ultrasonic liquid joining would be a reliable technique with excellent adaptability to different environments and joining forms.

In summary, we successfully joined MGs in different liquid environments by using simple and convenient ultrasonic vibration technology. The whole UVJ process exhibited low temperature and no current state, which will completely avoid the degradation of joined quality caused by high temperature and the unsafe problems caused by high current in common underwater connection techniques. In addition, the joined samples not only showed no obvious defects in the joined interface, but also exhibited excellent mechanical properties and corrosion resistance. Our approach not only presents an effective route to join underwater for offshore and marine application, but also provide a feasible strategy for joining under extreme environments like flammable environment in oil, gas, organic solvents and cryogenic conditions in space.

## Methods
### Sample preparation
The compositions of La-based ($La_{55}Al_{25}Ni_5Cu_{10}Co_5$), Zr-based ($Zr_{55}Cu_{30}Al_{10}Ni_5$) MGs and TiZrHfBeNi high entropy bulk metallic glass (HE-BMG) were selected as the joining materials for this experiment. The as cast materials were prepared by casting pure monomers (>99.99% purity) into bulks by conventional water-cooled copper molds under a low-pressure vacuum environment with inert gas argon. The as cast materials were cut into various sizes by using a low-speed diamond cutting for different joining types.

### Joining under ultrasonic vibration
Ultrasonic-related equipment is primarily composed of an ultrasonic generator (emits electrical signals), a control unit (setting parameters), and an ultrasonic vibration machine (converts electrical signals into vibration signals), as shown in Supplementary Fig. 8. The ultrasonic vibration device was used to join MGs with a vibration frequency of 20000 Hz (+-500 Hz) and a maximum power of 2500 W. The shape of

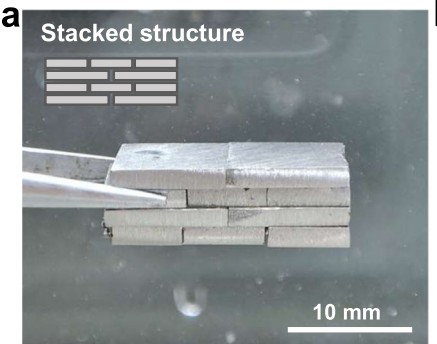

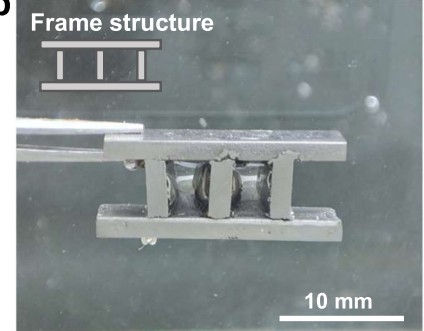

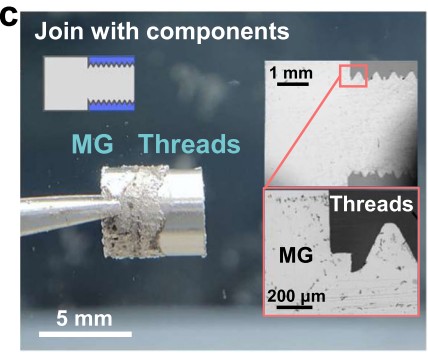

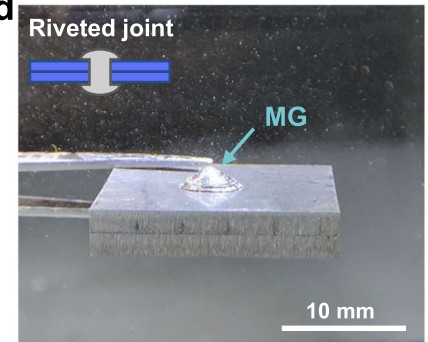

**Fig. 6 | Different complex joining types under liquids. a** Stacked structure. **b** Frame structure. **c** Joining with threads based on the flow deformation of MGs under ultrasonic vibration. The inset shows longitudinal section images of the sample, which represents the tight connection of the MG and the threaded structure. **d** Riveting of steel parts based on the ultrasonic softening properties of MGs.

the vibration in the sonotrode presents a sinusoidal function with an amplitude of 44.4 μm (+-1 μm) (Fig. 1a). In this UVJ experiment, the sonotrode comes in contact with the sample at an input trigger force (200 N) and then starts vibrating. In order to better control the effect of ultrasonic vibration to the sample, the energy mode was chosen as the abort (stop) criterion. In the joining process, the pressure was measured in real-time using a homemade force gauge, and a data acquisition card (National Instruments NI-9237) with a sampling frequency of 1 kHz was used to process and transfer the data to a computer. The high-speed camera (Revealer X150) was used to capture the joining process. Different types of joining (butt joint of cylindrical, butt joint of sheet, lap joint, T-joint) are realized by using different clamping apparatus. The clamping method of various joining types is shown in Supplementary Fig. 9.

### Temperature measurement

The temperature was measured in real-time using a K-type thermocouple, and a data acquisition card (National Instruments NI-9237) with a sampling frequency of 100 Hz was used to process and transfer the data to a computer, which can achieve an accuracy of 0.2° C. To capture the temperature more accurately, make the thermocouple measurement point located at the interface, as shown in Supplementary Fig. 10. After the test, to exclude the effect of heat transfer during the measurement, verify that the spilled MGs at the interface cover the thermocouple. An infrared imager (Fotric 280d) was used to detect thermal images of the joining process in air.

### Multiple characterization

The XRD (Rigaku miniflex600) with Cu-Kα radiation at a scanning rate of 5° min⁻¹ from 20° to 80° was used to verify the amorphous properties of the material. In order to measure the enthalpy change of MGs, differential scanning calorimetry (DSC 3, METTLER TOLEDO) was carried out at a heating rate of 15 K•min⁻¹ in a pure argon atmosphere. The morphology was captured using a field emission scanning electron

microscope (SEM) (FEI QUANTA FEG 450) equipped with energy-dispersive X-ray spectroscopy (EDS). The microscale interface structures were observed using the high angle annular dark field (HAADF) and the selected-area electron diffraction (SAED) images were obtained by a scanning transmission electron microscope (TEM, Fei Titan Themis). The composition distribution analysis at the joined interface was conducted using nanobeam energy-dispersive X-ray spectroscopy (EDX) with a beam diameter of 2 nm. The three-dimensional visual characterization of the joined sample was performed by the computed tomography (CT, Sanying precision instruments-nano Voxel 3000d) device. The dynamic scanning probe microscopy (DSPM) was measured by using two techniques: (1) The nano-DMA mode on a nanoindentation instrument (TI950, Hysitron) with an applied tip frequency of 200 Hz and (2) the AMFM (amplitude modulation-frequency modulation) viscoelastic mapping mode was implemented on an atomic force microscope (Cypher S AFM, Oxford Instrument) with a frequency of 72,000 Hz. To minimize the influence of surface roughness to the experimental results, all DSPM tests were conducted on $Zr_{55}Cu_{30}Al_{10}Ni_5$ MG-coated samples.

### Mechanical properties test

The hardness of the cross section was measured with a Vickers hardness tester under a load of 500 gf (4.90 N) and held for 10 s, and the dot pitches are 200 μm. To characterize the hardness and modulus of the joined samples and the as cast MGs, nanoindentation experiments were conducted using a Berkovich triangular pyramid indenter with a tip radius of 20 nm. In order to perform the tensile test accurately, the joined samples were cut into dog-bone-shaped with a thickness of 0.5 mm, gauge width of 1.5 mm, and gauge length of 4.8 mm samples (see Supplementary Fig. 11). Tensile tests were performed at room temperature with the strain rate of 0.001 s⁻¹. Three-point bending tests were performed on the butt joint (sheet) specimens with dimensions of 20 mm × 10 mm × 0.5 mm at a displacement velocity of 0.5 mm/min and a support span of 10 mm.

## Corrosion performance test

The 5 mm cylindrical joined specimens were cut in half, and their longitudinal-section was selected as the surfaces for corrosion testing, as shown in Supplementary Fig. 12. Before the corrosion test, the electrochemical polarization was performed in a three-electrode cell containing a platinum counter electrode and a saturated calomel reference electrode (SCE). The electrolyte used in this work was a 1 M HCl solution. Before the corrosion test, the specimens were mechanically polished to a mirror finish, then degreased in acetone, washed in distilled water, dried in air, and further exposed to air for 24 hours to obtain good reproducibility. The polarization curves of the specimen potentials were measured at a potential scan rate of 1 mV/s. In order to further observe the corrosion of the specimen surface, the specimen was anodized to the extent that pitting corrosion occurred and immediately removed for morphological examination under SEM.

## Data availability

All data needed to evaluate the conclusions in the paper are present in the paper and/or the supplementary information. Data are also available from the corresponding author upon request.

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

## Acknowledgements

J.M. was supported by the Key Basic and Applied Research Program of Guangdong Province, China (Grant Nr. 2019B030302010), the NSF of China (Grant Nr. 52122105, 51871157, 51971150), the National Key Research and Development Program of China (Grant No. 2018YFA0703604). The authors thank the assistance on microscope observation received from the Electron Microscope Center of Shenzhen University, and the nanoindentation instrument (TI950, Hysitron) test from Materials Growth and Characterization Center in Songshan Lake Materials Laboratory. The authors also appreciate the useful discussions with Prof. Wei-Hua Wang.

## Author contributions

L.L., Z. H. and J.M. conceived the work. J.M. and S.R. supervised the work. L.L. and X.L. conducted the ultrasonic experiments, and Z.H. designed the experimental setup. J.H. and J.F. performed the SEM. Z.L. and L.L. performed the corrosion performance test. W.W., Y.Z. and S.H. conducted the reference investigation. L.L. carried out TEM observation, XRD, CT and mechanical properties test. L.L. and J.M. wrote the manuscript. All authors contributed to the discussion and analyzed the results.

## Competing interests

The authors declare no competing interests.
