## [Peer Review File · Nature Communications]

Joining of metallic glasses in liquid via ultrasonic vibrationsREVIEWER COMMENTS

Reviewer #1 (Remarks to the Author):

The paper addresses an exciting and technologically relevant topic. I like the investigations about the fundamental bonding mechanism and the explanations of the mechanical and chemical properties of the under-liquid joints in comparison to the amorphous metal. The approach is innovative and exciting from my point of view.

Nevertheless, I have some questions and comments on the article:

- The motivated application for underwater (liquid) welding in the marine or space sector is understandable. Please explain in more detail and especially why you only consider metallic glasses and high-entropy alloys somehow. I also see some other potential material systems for those applications in harsh environment, such as Ni-, Co- or Fe-based alloys.
- the description of the ultrasound technique used is a bit too brief for me. Some information about the frequency (exactly 20,000 Hz) cannot be correct, because the resonance systems always depend on the acoustic properties of the components, i.e. we are talking about 20 KHz +- up to a few hundred hertz. The same applies to the vibration amplitudes. This is given here as 44.4 μm . These are definitely values that I would expect for the challenging welding task. Nevertheless, I am missing statements on the determination of the process parameters used. How were the specified values (200 N / 44.4 μm) determined? Furthermore, which abort (stop) criterion was chosen for the joints? There are different approaches possible in general (time, energy, distance/displacement).
- the biggest and most important question I have at the moment is regarding the direction of vibration applied and actually taking place at the interface. As shown in the principle sketches, the vibrations are applied perpendicular to the interface. As far as I know, this only leads (more or less) to the so-called hammering effect combined with a corresponding increase in temperature. This variant is used very successfully for bonding thermoplastics, so polymers. For metals with the corresponding oxide layers also mentioned in the paper, a vibration component parallel to the interface is mandatory to crack the layers and "clean" the reactive interface. In the additional images of the highspeed camera (very nice by the way), it is difficult for me to recognise these directions, but I can also capture parts of parallel movements, which are typical and causal for the bonding of metals, including with bulk metallic glasses, and which have already been published in many cases - although not under-liquid conditions for sure.
- the temperature measurements (Fig. 2) are interesting. I recommend to be a little more cautious about interpreting the measured temperatures. Where exactly the measurements were taken and what is about cooling and heat transfer in the under-liquid conditions, are those effects taken into account?
- please explain more about your idea on the "hypothetical active atoms" at the interface. Furthermore please add more on your statement about the activated "fluid-like behaviour" of the metallic glass.

- mechanical performance: besides tensile stress, bending in particular is the most important load case for engineering structures. It would be nice to show corresponding bending properties if available.
- The corrosion work is very exciting and interesting. Please add the reasons for the improved properties in the interface. Is there a possibility to develop the ultrasonic effect in terms of processing - in addition to the possibility of bonding?

- Please add - similar to other areas - the information on the central equipment of your investigations, so all the used ultrasonic components (see Suppl. Fig. 4). In this context, please also explain what exactly you mean by the phrase "The mechanism is very different compared to other ultrasonic-related treatments". I see here approaches for welding of polymers at 20 kHz as well as aspects of ultrasonic cleaning and thus their combination. Essential changes should be described in your paper sufficiently as well as precise as possible.

Reviewer #2 (Remarks to the Author):

This is the review comments of the paper entitled "Joining of metallic glasses in liquid via ultrasonic vibrations" by L.Y. Li et al. This paper includes some interesting results on the success of joining of two BMGs (ZrCuAlNi and high entropy BMG). However, the details of the joining process and the joining mechanism are not clear and there are many unknown and unsolved points. In addition, a number of previous papers on the welding of BMGs have not been cited. The unknown and unclear points are as follows:

1. The references on the general welding methods for metallic crystalline materials have been cited, but any previous papers on bulk metallic glasses (BMGs) by the other authors have not been cited. There are some previous papers on various welding methods including ultrasonic vibration welding for BMGs. The authors should cite previously reported many papers on welding for BMGs.
2. The temperature rise during ultrasonic vibration joining shown in Fig. 2 is very low (75-98 °C), being significantly different from the actual temperature rise in the really vibration welded part. The welding temperature is very important for the understanding of the welding mechanism during ultrasonic vibration joining. The temperature measurement method during the welding should be improved.
3. The temperatures of 88.79°C, 98.04°C, 93.98°C and 74.93°C have been presented. It is difficult to measure such high accuracy of the temperature. The extremely high accurate temperature measurement way should be presented.
4. The color word "blue" seems to be "green".
5. The interface between ZrCuAlNi BMG and high entropy type BMG is not smooth and has an irregular interface morphology. The mechanism for the generation of such an irregular interface should be presented.
6. The glass transition temperatures of the above-described two BMGs are different significantly. Based on the difference in the glass transition temperature, the joining mechanism for the two BMGs should be explained.
7. The ultrasonic vibration joining is thought to occur through atomic mobility. The influence of ultrasonic vibration at the frequency of 20000 Hz on the atomic mobility and atomic diffusion should be presented.
8. In the case of ultrasonic energy (700 J), the mechanism for the good joining should be explained in more details.
9. The influence of the oxide layer on the joined surface is not fully explained. The details on the decomposition behavior of oxide phases such as ZrO₂ etc. should be presented.

10. How much is the oxygen content at the interface between the ZrCuAlNi BMG and the high entropy BMG?

10. The uniform distribution mechanism of oxygen at the low temperature into the matrix part should be presented. The diffusivity of oxygen to the matrix phase at low temperatures is very low and hence the homogeneous distribution of oxygen from ZrO₂ oxide seems to be difficult.

11. The more detailed information on the joined interface structure should be presented. The authors have claimed that the joining occurs through atomic diffusion. However, there are no information on the diffusion distance and on the possibility of the formation of a new BMG phase region with different composition. Even in the high-resolution TEM image, there are no data on new image contrast region. The authors should present the change in alloy composition distribution at the joined interface by nanobeam EDX analytical method. Some previous papers have presented such finely analyzed data at the joined interface part.

12. The tensile test was made for the thin sheet with a thickness of 0.5 mm. Is it possible to measure the tensile testing data for the thicker sheet sample?

13. There is no explanation on the mutual relation between the tensile fracture surface (site) and the joined interface part. The clarification on this relation is extremely important for the engineering usefulness of this welding process.

14. According to some previous papers on the fractured surface for the joined BMGs, the joined interface in the two BMGs has been observed clearly in the fracture surface SEM images, being completely different from the present fracture surface image. The reason for the inconsistent data on the fracture surface should be presented.

15. The details for the specimen part used for the measurement of corrosion behavior should be presented.

16. The difference in the corrosion resistance seems to be recognized among the three parts, namely, ZrCuAlNi BMG, joined part and high entropy BMG. The difference in the corrosion resistance among the three phase regions should be presented.

17. According to many previous data on the corrosion resistance, the corrosion resistance of BMGs decreases upon annealing. The present data appears to be opposite. The inconsistent reason should be explained.

18. The similar data have already been presented in Scripta Mater. in 2020. The authors should clarify the difference between the previous data and the present data. This manuscript seems to be a kind of technical report. The more detailed interpretation and discussion should be required to improve the quality of this manuscript.

Based on the above-described comments, it is regrettable to conclude that the present manuscript should be rejected for publication of Nature Communication.

Reviewer #3: (Remarks to the Author):

This paper reports an ultrasonic vibration based welding of metallic glass in various liquids. What they showed in the welding demos as well as the comparable strength of welded sample with the base alloy are very impressive. The experiments are in general vigorously done and the characterization are sound. It may be appropriate to publish in Nature Communications. I have several questions, which are described in the following:

1. It seems that the ultrasonic vibration welding is very suitable for liquid environment welding, it is suggested to add discussion on the challenge of ultrasonic vibration based underwater welding by comparison with the traditional ultrasonic vibration welding in air.
2. Is there any difference in the welding mechanism by using ultrasonic vibration welding in air and in liquid? It is suggested to compare the welding performance of joined samples in air and in different liquids (pure water, seawater, alcohol and liquid nitrogen), but with the same ultrasonic vibration welding conditions.
3. The Referee is very interested in the welding mechanism, the authors attributed it to the breakage of oxide layer and atom diffusion. It is hard to imagine that the oxide layer can break into atomic scale size (700 J in Fig. 3b, referring to such as: *Acta Mater.*, 62, 49-57, 2014), is it possible that just atoms diffusion into oxide layer and significantly dilute the oxygen content the oxide layer?
4. The authors attribute the good welding performance to the fast surface dynamics induced fluid-like behavior, what's the specific surface dynamic behavior (liquid or BMG related?) and underlying mechanism? Is it the unique behavior of ultrasonic vibration welding in liquid or not? More evidence and details are needed.
5. Fig. 3 clearly shows the effect of oxide layer on the welding performance. I am wondering that is ultrasonic welding performance better in liquid environment with less oxygen content than in air? If so, is the required threshold input energy (for example, 700 J in Fig. 3a-b) also lower in liquid than in air?
6. The authors attributed the honeycomb-like structure in the non-interface area (Fig. 4j) to intense corrosion, but it seems more like the vein-pattern of typical fractured BMG surface to the Referee (referring to such as: *Acta Mater.*, 62, 49-57, 2014), I think this is important to the mechanism understanding and suggest the authors to clarify it.

Responses to Reviewers' Reports on NCOMMS-22-50999

We would like to thank the referees for reviewing our manuscript entitled “Joining of metallic glasses in liquid via ultrasonic vibrations” (NCOMMS-22-50999). We are grateful that all the three reviewers raised positive comments on this work, and provided constructive suggestions to help us to substantially improve the quality of this work.

In the following, we provide below a point-to-point reply to all comments of the referees. All the concerns have been properly addressed, we also included the comments below followed by our responses in blue for your convenience.

Response to Reviewer #1

The paper addresses an exciting and technologically relevant topic. I like the investigations about the fundamental bonding mechanism and the explanations of the mechanical and chemical properties of the under-liquid joints in comparison to the amorphous metal. The approach is innovative and exciting from my point of view.

Nevertheless, I have some questions and comments on the article:

Response:

We thank the reviewer for the positive comments.

1. The motivated application for underwater (liquid) welding in the marine or space sector is understandable. Please explain in more detail and especially why you only consider metallic glasses and high-entropy alloys somehow. I also see some other potential material systems for those applications in harsh environment, such as Ni-, Co- or Fe-based alloys.

Response:

Thank you for your professional and valuable concern. The materials used in present work are metallic glasses (MGs), which are amorphous in atomic packing. Different from the conventional crystalline alloys, MGs show the unique ultrasonic vibrations induced plasticity makes them flow like the viscous liquid. As a result, the

effective joining under ultrasonic vibrations can be achieved at low temperatures even under liquid can be achieved. In contrast, the crystalline alloys such as Ni-, Co- or Fe-based alloys do not have such property, therefore, we did not realize the under-liquid joining in these materials.

2. the description of the ultrasound technique used is a bit too brief for me. Some information about the frequency (exactly 20,000 Hz) cannot be correct, because the resonance systems always depend on the acoustic properties of the components, i.e. we are talking about 20 KHz +- up to a few hundred hertz. The same applies to the vibration amplitudes. This is given here as 44.4 μm . These are definitely values that I would expect for the challenging welding task. Nevertheless, I am missing statements on the determination of the process parameters used. How were the specified values (200 N / 44.4 μm) determined? Furthermore, which abort (stop) criterion was chosen for the joints? There are different approaches possible in general (time, energy, distance/displacement).

Response:

Thank you for your professional concern.

In fact, the frequency of 20,000 Hz is a preset value of the ultrasonic device. The frequency error range listed in the ultrasonic device manual is 20,000 Hz (+-500 Hz), and the device has its own actual frequency self-test system showing 20,106 Hz (**Fig. R1-1**). Therefore, the frequency of 20,000 Hz is not an exact value

The amplitude is also an input value, in our experiments we used the amplitude rate of 100%. In order to investigate the actual amplitude of the device, we measured the actual amplitude using a laser vibrometer. According to the test results, we can obtain the actual output amplitude is 44.4 μm (+-1 μm) (**Fig. R1-2**). Similar to the amplitude, the triggering pressure of ultrasonic vibration is also an input parameter, and it is inevitable that there are deviations in the actual output values. We chose 200 N/44.4 μm as the input value because several preliminary experiments showed that this combination of parameters yields a relatively superior joining performance.

The ultrasonic vibration device we used have three abort (stop) criterions: energy,

time, and relative/absolute displacement modes. In order to better control the effect of ultrasonic vibration to the sample, the energy mode was chosen in this work.

Fig. R1-1. Frequency self-test system for ultrasonic equipment.

Fig. R1-2. The displacement of the sonotrode during the ultrasonic vibration detected by the laser vibrometer.

Changes made in the revised manuscript:

(1) We have **added** the displacement diagram of the sonotrode in Fig. 1(a).

(2) Page 21, Line 449:

We have **added** more detailed and accurate descriptions of frequency, amplitude and bort (stop) criterion in Methods.

3. the biggest and most important question I have at the moment is regarding the

direction of vibration applied and actually taking place at the interface. As shown in the principle sketches, the vibrations are applied perpendicular to the interface. As far as I know, this only leads (more or less) to the so-called hammering effect combined with a corresponding increase in temperature. This variant is used very successfully for bonding thermoplastics, so polymers. For metals with the corresponding oxide layers also mentioned in the paper, a vibration component parallel to the interface is mandatory to crack the layers and “clean” the reactive interface. In the additional images of the highspeed camera (very nice by the way), it is difficult for me to recognise these directions, but I can also capture parts of parallel movements, which are typical and causal for the bonding of metals, including with bulk metallic glasses, and which have already been published in many cases - although not under-liquid conditions for sure.

Response:

Thank you for your valuable concern.

In our work, the vibrations are applied perpendicular to the interface of the sample, which makes MGs soften due to the dynamic heterogeneity of MGs and the expansion of the liquid-like region induced by high-frequency cyclic stress¹⁻³, instead of the thermal mechanisms induced by parallel vibration (can be ascertained by the temperature detection in **Fig R1-4** and structure measurement by DSC in Figure 5m). The high-speed cameras also captured the softening of MGs process without obvious parallel vibration (see **Response Movie R1**). There is the fundamental difference from the softening or bonding of other metals in the air, which is based on thermal mechanisms. Owing to the non-thermal softening of MGs under ultrasonic vibrations, the joining of them can be realized.

Despite the softening of MGs was not caused by the parallel vibration, we think it is difficult to ensure that the sample does not move in parallel at all due to the lack of tight constraints and the unevenness of the MG surface. After the softening of MGs under perpendicular vibrations, these possible relative parallel movements maybe facilitate the crack of the oxide layers and “clean” the reactive interface, as stated by the reviewer.

- 1 Qiao, J. *et al.* Structural heterogeneities and mechanical behavior of amorphous alloys. *Progress in Materials Science* **104**, 250-329 (2019).
- 2 Galindo-Torres, S., Zhang, X. & Krabbenhoft, K. Micromechanics of liquefaction in granular materials. *Physical Review Applied* **10**, 064017 (2018).
- 3 Li, X. *et al.* Ultrasonic plasticity of metallic glass near room temperature. *Applied Materials Today* **21**, 100866 (2020).

Changes made in the revised manuscript:

(1) We have **added** the direction of vibration in Fig. 2(a).

(2) Page 13, Line 272:

We have **revised** a description about ultrasonic-induced softening mechanism of MG and the oxide layer fragmentation mechanism in **Results and discussion**.

4. the temperature measurements (Fig. 2) are interesting. I recommend to be a little more cautious about interpreting the measured temperatures. Where exactly the measurements were taken and what is about cooling and heat transfer in the under-liquid conditions, are those effects taken into account?

Response:

Thank you for your valuable suggestion.

Frist, our temperature test system can achieve an accuracy of 0.2°C and a sampling frequency of 100 Hz, which is sufficiently accurate. This temperature testing system includes a K-type thermocouple (TT-K-36), a data acquisition card (National Instruments NI-9237), and temperature measurement software installed on a computer.

Secondly, to be more precise, we re-designed the temperature measurement structure and re-measured the temperature. We positioned the temperature sensors as close as possible to the interface and created three micro-slots on the MG interface (**Fig. R1-3**). The thermocouples were secured with adhesive and contacted with the MGs samples closely, without the cooling and heat transfer influence. After the test and join process was completed, the thermocouples and micro-grooves were covered by spilled

softened MG, which verified the reliability of the temperature data (**Fig. R1-3b**). The re-measured temperature data is presented in the **Fig. R1-4**. The results are similar to those presented in the original manuscript.

Fig. R1-3. a, Schematic diagram of temperature measurement method. **b**, physical display of the temperature measurement process.

Fig. R1-4. The temperature profiles of joining in 4 types of liquid environments detected by thermocouples.

Changes made in the revised manuscript:

- (1) We have **revised** the temperature profiles in Fig. 2(b) and Supplementary Figure 2.
- (2) We have **added** Fig R1-3 as Supplementary Figure 10.
- (3) Page 22, Line 469:

We have **added** the temperature measurement details in Methods.

(4) Page 8, Line 150

We **revised** the description of temperature measurement result in Results and discussion.

5. please explain more about your idea on the "hypothetical active atoms" at the interface. Furthermore please add more on your statement about the activated "fluid-like behaviour" of the metallic glass.

Response:

Thank you for the suggestion. We have to admit that the "hypothetical active atoms" is an inappropriate term that we used to express schematically the active atomic diffusion at the interface. Therefore, we have made a descriptive change as "the red and yellow spheres in Fig. 3k schematically represent the diffusion of atoms at the two-MGs interface".

We added a description to explain the "fluid-like behavior" mechanism of metallic glass in the revised manuscript: The MGs consist of liquid-like and solid-like regions, a phenomenon known as structural heterogeneity¹. The solid-like regions form an elastic network through percolation, while the liquid-like regions enclosed in the network act as viscous flow units that dissipate energy. Under cyclic loading, the "liquid-like" regions bear larger pressure. Due to the absence of sufficient time for stress relaxation, the pressure accumulates, causing the expansion of the liquid-like regions and the collapse of the entire amorphous structure^{2,3} (**Fig. R1-5**). As shown in the dynamic scanning probe microscope (DSPM) image of the $Zr_{55}Cu_{30}Al_{10}Ni_5$ MG, therein the viscoelastic loss tangent ($\tan\delta$) is proven to correspond to internal friction. When the frequency increase to the ultrasonic vibrations, the $\tan\delta$ value increases rapidly and all regions exhibit higher than 0.22 (**Fig. R1-6a-c**), indicating the activation and extensive expansion of the liquid-like regions. In order to express the "fluid-like behavior" of MG more intuitively, the viscosity was calculated using the Maxwell model and it was demonstrated that the viscosity decreases by three orders of magnitude under ultrasonic vibration (**Fig. R1-6d**). We have added the above discussion in the revised manuscript.

Fig. R1-5. Schematic diagram of ultrasonic vibration-induced softening and joining of the interface of MG (atomic scale).

Fig. R1-6. **a, b.** The viscoelastic loss tangent map at $f = 200$ and $72,000$ Hz. **c,** The statistical analysis of **b** and **c** with a good fit of Gaussian distribution. **e,** Viscosity (or relaxation time) distribution after normalizing the value of the peak position at $f = 200$ Hz. **d,** Schematic diagram of the mechanism for UVJ with irregular interface morphology.

- 1 Qiao, J. *et al.* Structural heterogeneities and mechanical behavior of amorphous alloys. *Progress in Materials Science* **104**, 250-329 (2019).
- 2 Galindo-Torres, S., Zhang, X. & Krabbenhoft, K. Micromechanics of liquefaction in granular materials. *Physical Review Applied* **10**, 064017 (2018).
- 3 Li, X. *et al.* Ultrasonic plasticity of metallic glass near room temperature. *Applied Materials Today* **21**, 100866 (2020).

Changes made in the revised manuscript:

(1) Page 12, Line 244

We have **modified** the description about “hypothetical active atoms”: “The red and yellow spheres in Fig. 3k schematically represent the diffusion of atoms at the two-MGs interface.”.

(2) Page 13, Line 272

We have **added** ultrasonic-induced softening mechanism in Results and discussion.

(3) We have **added** Fig R1-5 and Fig R1-6 in Fig.4.

6. mechanical performance: besides tensile stress, bending in particular is the most important load case for engineering structures. It would be nice to show corresponding bending properties if available.

Response:

We appreciate the important suggestion from the reviewer. We conducted bending tests on the butt-jointed (sheet) samples that were ground to a size of $20 \times 10 \times 0.5$ mm. The results showed that the three-point bending strength of the joined samples was close to that of the as-cast samples, as shown in **Fig R1-7**.

Fig. R1-7. Three-point bending engineering stress–strain curves of as cast Zr-based MG and joined sample.

Changes made in the revised manuscript:

(1) Page 17, Line 252

We have **added** the description about bending tests result.

(2) Page 24, Line 512

We have **added** the description about bending test detail in Methods.

(3) We have **added** Fig R1-7 in Fig.5.

7. The corrosion work is very exciting and interesting. Please add the reasons for the improved properties in the interface. Is there a possibility to develop the ultrasonic effect in terms of processing - in addition to the possibility of bonding?

Response:

We thank the reviewer for this positive comment. Previous work have proved that the corrosion resistance of MG is closely related to its amorphous structural or energy state⁴. In the past years, a large number of studies to optimize the corrosion resistance through relaxation of MG have been reported⁴⁻⁶. That is due to the fact that the relaxation phenomenon reduces the average atomic spacing and chemical potential of MGs, allowing them to reach a more stable state, which will greatly improve corrosion resistance. After ultrasonic vibration, we have studied the interface and found that quite significant relaxation occurred at the interface compared to the cast MG and the non-interface area of the joined sample (**Fig R1-8a**). Therefore, the interface exhibits improved properties. We have added the relevant discussion in the revised manuscript.

In fact, we have currently demonstrated the processing to enhance the corrosion resistance of MG by ultrasonic vibration, and preliminary results was obtained, as shown in **Fig.R1-8a**.

Fig. R1-8. (a) DSC traces of the close-up view of the relaxation exotherm. (b) the preliminary results about the processing to enhance the corrosion resistance of MG.

- 4 Jiang, W. et al. Electrochemical corrosion behavior of a Zr-based bulk-metallic glass. *Appl. Phys. Lett.* **91**, 041904 (2007).
- 5 Jindal, R. *et al.* Effect of annealing below the crystallization temperature on the corrosion behavior of Al–Ni–Y metallic glasses. *Corros. Sci.* **84**, 54-65 (2014).
- 6 Zhou, M., Hagos, K., Huang, H., Yang, M. & Ma, L. Improved mechanical properties and pitting corrosion resistance of $Zr_{65}Cu_{17.5}Fe_{10}Al_{7.5}$ bulk metallic glass by isothermal annealing. *J. Non-cryst. solids* **452**, 50-56 (2016).

Changes made in the revised manuscript:

(1) Page 18, Line 391

We have **revised** the description for the improved corrosion properties.

8. Please add - similar to other areas - the information on the central equipment of your investigations, so all the used ultrasonic components (see Suppl. Fig. 4). In this context, please also explain what exactly you mean by the phrase "The mechanism is very different compared to other ultrasonic-related treatments". I see here approaches for welding of polymers at 20 kHz as well as aspects of ultrasonic cleaning and thus their combination. Essential changes should be described in your paper sufficiently as well as precise as possible.

Response:

Thank you for your professional concern. We have re-photographed a more complete picture of the equipment and added detailed descriptions

The phrase "The mechanism is very different compared to other ultrasound-related therapies" mean to the fact that our ultrasonic vibrations are different from ultrasonic-wave (including ultrasonic cleaning). Our ultrasonic vibration is a high frequency mechanical loading, which means the sonotrode has a direct contact with the sample,

while there is only an ultrasonic wave field in the traditional ultrasonic cleaning, such as glass cleaning. To prevent misunderstanding, we have deleted this sentence.

Fig. R1-8. Equipment diagrams.

Changes made in the revised manuscript:

(1) We have **added** Fig R1-8 as the Supplementary Figure 8.

(2) Page 21, Line 449

We have **added** description of central equipment in Methods.

(3) Page 21, Line 445

We have **deleted** the sentence “The mechanism is very different compared to other ultrasonic-related treatments”.

Response to Reviewer #2

This is the review comments of the paper entitled “Joining of metallic glasses in liquid via ultrasonic vibrations” by L.Y. Li et al. This paper includes some interesting results on the success of joining of two BMGs (ZrCuAlNi and high entropy BMG). However, the details of the joining process and the joining mechanism are not clear and there are

many unknown and unsolved points. In addition, a number of previous papers on the welding of BMGs have not been cited. The unknown and unclear points are as follows:

Response:

We are glad that the reviewer thinks our results are interesting and we also thank the reviewer for the valuable comments on the joining process and mechanisms. We have made significant revisions on the manuscript, which may properly address the concerns of the reviewer.

1. The references on the general welding methods for metallic crystalline materials have been cited, but any previous papers on bulk metallic glasses (BMGs) by the other authors have not been cited. There are some previous papers on various welding methods including ultrasonic vibration welding for BMGs. The authors should cite previously reported many papers on welding for BMGs.

Response:

Thank you for your valuable suggestion. We have cited the related literature about the various welding methods of BMGs in the revised manuscript.

Changes made in the revised manuscript:

(1) Page 5, Line 79

We have **added** descriptions and cited the literature about various welding methods of BMGs in the "Introduction" section: "At present, there are two main mechanisms for joining MG. One involves melting MG and subsequent rapid quenching, mainly including laser welding³³, explosion welding³⁴ and electric pulse welding³⁵. Another potential method is joining in the supercooled liquid region, such as thermoplastic joining³⁶, friction stir welding³⁷. These methods also require extremely high heat input and complex devices, making them difficult to apply in harsh liquid environments. Therefore, up to now, there is no one technology that can truly implement MG-MG joining under liquid. In recent years,

a novel method that can achieve joining MGs at low temperatures has emerged - ultrasonic vibration joining (UVJ) technology^{38,39}. Due to its low temperature requirement and simple operation, UVJ technology has great potential for joining MG in liquid environments”.

The references include:

- 33 Wang, G., Huang, Y. J., Shagiev, M. & Shen, J. Laser welding of Ti₄₀Zr₂₅Ni₃Cu₁₂Be₂₀ bulk metallic glass. *Materials Science and Engineering: A* **541**, 33-37, (2012).
- 34 Liu, K. X. *et al.* Atomic-scale bonding of bulk metallic glass to crystalline aluminum. *Applied Physics Letters* **93**, doi:10.1063/1.2976667 (2008).
- 35 Fujiwara, K., Fukumoto, S., Yokoyama, Y., Nishijima, M. & Yamamoto, A. Weldability of Zr₅₀Cu₃₀Al₁₀Ni₁₀ bulk glassy alloy by small-scale resistance spot welding. *Materials Science and Engineering: A* **498**, 302-307, (2008).
- 36 Chen, W., Liu, Z. & Schroers, J. Joining of bulk metallic glasses in air. *Acta Materialia* **62**, 49-57, (2014).
- 37 Wang, D., Xiao, B. L., Ma, Z. Y. & Zhang, H. F. Friction stir welding of Zr₅₅Cu₃₀Al₁₀Ni₅ bulk metallic glass to Al–Zn–Mg–Cu alloy. *Scripta Materialia* **60**, 112-115, (2009).
- 38 Ma, J. *et al.* Fast surface dynamics enabled cold joining of metallic glasses. *Science advances* **5**, eaax7256 (2019).
- 39 Li, X., Liang, X., Zhang, Z., Ma, J. & Shen, J. Cold joining to fabricate large size metallic glasses by the ultrasonic vibrations. *Scripta Materialia* **185**, 100-104 (2020).

2. The temperature rise during ultrasonic vibration joining shown in Fig. 2 is very low (75-98°C), being significantly different from the actual temperature rise in the really vibration welded part. The welding temperature is very important for the understanding of the welding mechanism during ultrasonic vibration joining. The temperature measurement method during the welding should be improved.

Response:

Thank you for your professional suggestions. We agree that the temperature measurement is very important for present work. According to you suggestion, we have

improved the temperature measurement method in the revised manuscript. We employed a high-precision temperature measurement system, with an accuracy of 0.2°C and a sampling frequency of 100 Hz.

We positioned the temperature sensors as close as possible to the interface and created three micro-slots on the MG interface (**Fig. R2-1**). The thermocouples were secured with adhesive and contacted with the MGs samples closely, without the cooling and heat transfer influence. After the test and join process was completed, the thermocouples and micro-grooves were covered by spilled softened MG, which verified the reliability of the temperature data (**Fig. R2-1b**). The re-measured temperature data is presented in the **Fig. R2-2**. The results are similar to those presented in the original manuscript.

However, we know the glass transition temperature of $\text{Zr}_{55}\text{Cu}_{30}\text{Al}_{10}\text{Ni}_5$ MG is about 390°C , which is much higher than the measured temperature. Therefore, the joining mechanism is not based on temperature rise.

Fig. R2-1. a, Schematic diagram of temperature measurement method. b, Physical display of the temperature measurement process.

Fig. R2-2. The temperature-time curve of Zr-based join measured by the thermocouple.

Changes made in the revised manuscript:

- (1) We have **revised** the temperature profiles in Fig. 2(b) and Supplementary Figure 2.
- (2) We have **added** Fig R2-1 as Supplementary Figure 10.
- (3) Page 22, Line 469:

We have **added** temperature measurement details in Methods.

- (4) Page 8, Line 150

We **revised** the description of temperature measurement result in Results and discussion.

3. The temperatures of 88.79°C, 98.04°C, 93.98°C and 74.93°C have been presented. It is difficult to measure such high accuracy of the temperature. The extremely high accurate temperature measurement way should be presented.

Response:

Thank you for your suggestions. Our temperature testing system includes a K-type thermocouple (TT-K-36), a data acquisition card (National Instruments NI-9237), and temperature measurement software installed on a computer. After research the device manual, we found that our homemade temperature testing system can achieve an

accuracy of 0.2°C. We identified that the temperature measurement data displayed with a precision of two decimal places was an artifact generated by the system rather than the reflection of the actual measurement accuracy. We have corrected this error in the data analysis and revised the temperature data to one decimal place.

Changes made in the revised manuscript:

(1) We have **modified** the temperature values in revised manuscript.

4. The color word “blue” seems to be “green”.

Response:

Thank you for your comments. We have modified the color word “blue” to “green”.

Changes made in the revised manuscript:

(1) Page 9, Line 172:

We have **modified** the color word ‘blue’ to ‘green’.

5. The interface between ZrCuAlNi BMG and high entropy type BMG is not smooth and has an irregular interface morphology. The mechanism for the generation of such an irregular interface should be presented.

Response:

Thank you for your valuable advice. Under the influence of ultrasonic vibration, the behavior of the fluidlike behavior is triggered by the dynamic heterogeneity of the MG and the expansion of the liquid-like region, resulting in the entire interface being in a fluid state. According to references^{7,8}, when high-intensity ultrasonic vibration acting on liquid, nonlinear effects such as cavitation and acoustic streaming will be triggered, resulting in high-intensity shock waves, strong convection, and turbulence. These effects promote the fragmentation and dispersion of the brittle oxide layer. On the other hand, the ultrasonic-induced flow provides strong mixing of two fresh liquid-like MGs, which promotes full contact of the interface and generates an irregular

interface. Therefore, the irregular interface can be formed. The whole process is shown in **Fig. R2-3**. According to the suggestion of the reviewer, we have added the mechanism in the revised manuscript.

Fig. R2-3. Schematic diagram illustrating the mechanism of oxide layer dispersion and interfacial bonding.

7 Eskin, G. (Moscow, Russia-Gordon and Breach science publisher, 1998).

8 Cui, Y., Xu, C. & Han, Q. Effect of ultrasonic vibration on unmixed zone formation.

Scripta materialia **55**, 975-978 (2006).

Changes made in the revised manuscript:

(1) Page 13, Line 272

We have **added** ultrasonic-induced softening mechanism in Results and discussion.

(2) Page 15, Line 313

We have **added** the mechanism for the generation of such an irregular interface in Results and discussion.

(3) We have **added** Fig R2-3 into Fig. 4.

6. The glass transition temperatures of the above-described two BMGs are different significantly. Based on the difference in the glass transition temperature, the joining mechanism for the two BMGs should be explained.

Response:

Thank you for your valuable advice. We checked the glass transition temperature of the two BMGs, and found that the ZrCuAlNi BMG is about 390°C (Figure 5m), while the glass transition temperature of the high-entropy type BMG is 388°C⁹, indicating little difference.

Most of all, in this work, the mechanism of joining is based on the low-temperature softening effect of MG under ultrasonic vibration, rather than the temperature raise effect. Under the influence of ultrasonic vibration, the behavior of the fluidlike behavior is triggered by the dynamic heterogeneity of the MG and the expansion of the liquid-like region, resulting in the entire interface being in a fluid state. The expansion of the liquid-like region is caused by the generation of high pressure around the "liquid-like" region due to cyclic loading. In the absence of sufficient time for stress relaxation, the pressure continues to accumulate, leading to the expansion of the liquid-like region^{2,3}.

After the MG near the interface is completely softened, high-intensity shock waves, strong convection, and turbulence will be triggered^{7,8}. These effects promote the fragmentation and dispersion of the brittle oxide layer. In addition, the ultrasonic-induced flow provides strong mixing of two fresh liquid-like MGs, which promotes full contact of the interface and generates an irregular interface. We have added the joining mechanism into the revised manuscript.

2 Galindo-Torres, S., Zhang, X. & Krabbenhoft, K. Micromechanics of liquefaction in granular materials. *Physical Review Applied* **10**, 064017 (2018).

3 Li, X. *et al.* Ultrasonic plasticity of metallic glass near room temperature. *Applied Materials Today* **21**, 100866 (2020).

7 Eskin, G. (Moscow, Russia-Gordon and Breach science publisher, 1998).

8 Cui, Y., Xu, C. & Han, Q. Effect of ultrasonic vibration on unmixed zone formation.

9 Gong, P., Li, F., Deng, L., Wang, X. & Jin, J. Research on nano-scratching behavior of

TiZrHfBeCu(Ni) high entropy bulk metallic glasses. *Journal of Alloys and Compounds* **817**, 153240 (2020).

Changes made in the revised manuscript:

(1) Page 13, Line 272

We have **added** ultrasonic-induced softening mechanism in Results and discussion.

(2) Page 15, Line 313

We have **added** the joining mechanism for the two BMGs.

7. The ultrasonic vibration joining is thought to occur through atomic mobility. The influence of ultrasonic vibration at the frequency of 20000 Hz on the atomic mobility and atomic diffusion should be presented.

Response:

Thank you for your professional concern. We measured the viscoelastic loss tangent map (**Fig R2-4b-c**) obtained by dynamic scanning probe microscopy (DSPM), which compared the changes in $\tan\delta$ at 200 Hz and 72,000 Hz (**Fig R2-4d**). Due to the fact that in the atomic force microscope (Cypher S AFM, Oxford Instrument) AMFM viscoelastic mapping mode, only 72,000 Hz and 13,200 Hz are available as choices (no 20000 Hz for selection). Therefore, here, we have chosen the 72,000 Hz (which is the same order of magnitude with our experiments) to characterize the effect of high frequency vibrations.

The literature indicates that $\tan\delta$ can generally correspond to internal friction¹⁰, similar to the results obtained from conventional dynamic mechanical analysis (DMA). In fact, the atomic migration rate has also been shown to be closely related to the internal friction ($\tan\delta$), with the atomic mobility increasing as the internal friction increases¹¹. It is found that the $\tan\delta$ is as high as 0.229 under the ultrasonic vibration of 72,000 Hz, while the $\tan\delta$ is only 0.130 under a low frequency. Such a high internal friction ($\tan\delta = 0.229$) is already close to the internal friction exhibited by BMGs after crossing the glass transition point¹², so quick atomic movement and atomic diffusion

must exist^{11,13}. Therefore, in this work, liquid-like MGs are easy to inter-diffuse, as shown in **Fig R2-5**, lots of diffusion layers was observed in joined interface. We have added the above discussion into the revised manuscript.

Fig. R2-4. **a**, Schematic diagram of ultrasonic vibration-induced softening and joining of the interface of MG (atomic scale). **b**, **c**. The viscoelastic loss tangent map at $f = 200$ and 72,000 Hz. **d**, The statistical analysis of **b** and **c** with a good fit of Gaussian distribution.

Fig. R2-5. **a**, **b**. The high angle annular dark field (HAADF) images at the interface of heterogeneous join. **c**. The nano-beam EDS through the diffusion layer. The scanning point is shown in **b**.

(2012).

- 12 Wang, W. H. The elastic properties, elastic models and elastic perspectives of metallic glasses. *Progress in Materials Science* **57**, 487-656 (2012).
- 13 Geyer, U. *et al.* Atomic diffusion in the supercooled liquid and glassy states of the Zr_{41.2}Ti_{13.8}Cu_{12.5}Ni₁₀Be_{22.5} alloy. *Physical review letters* **75**, 2364 (1995).

Changes made in the revised manuscript:

- (1) Page 14, Line 289

We have **added** the related content of dynamic scanning probe microscopy (DSPM) in Results and discussion.

- (2) We have **added** Fig R2-4 into Fig. 4 and provided a detailed description.
- (3) We have **added** Fig R2-5 into Fig. 3 and provided a detailed description.
- (4) We have **added** the related content about dynamic scanning probe microscopy (DSPM) in Methods.

8. In the case of ultrasonic energy (700 J), the mechanism for the good joining should be explained in more details.

Response:

Thank you for your professional concern. In this work, the mechanism of joining is based on the fluidlike behavior of MG under ultrasonic vibration. The behavior of the fluidlike behavior is triggered by the dynamic heterogeneity of the MG and the expansion of the liquid-like region under the influence of ultrasonic vibration, resulting in the entire interface being in a fluid state.

The effect of MG softening is related to the input energy. As the energy increases, the ultrasonic softening effect at the interface will be more fully realized. Only with sufficient softening behavior of MG can ultrasonic vibration effectively induce shock waves, strong convection, and turbulence in fluids, as shown in **Fig R2-6**. Under enough energy (700 J), the oxide layer can be more thoroughly broken, and mixing between the two phases can be more intense, resulting a more tightly bonding. As

shown in **Fig R2-7**, better bonding performance can be achieved when input energy increases. According to the suggestion, we have added the mechanism in the revised manuscript.

Fig. R2-6. Schematic diagram illustrating the mechanism of oxide layer dispersion and interfacial bonding.

Fig. R2-7. The joining performance with energy of 500 J and 700 J under water.

Changes made in the revised manuscript:

(1) Page 15, Line 313

We have **added** the mechanism for the good joining in Results and discussion,

“Under enough energy, after the MG near the interface is completely softened, high-intensity ultrasonic vibration applied to liquids can generate cavitation and acoustic streaming effects.....”

- (2) We added Fig. R2-6 into the revised manuscript as Fig. 4 and provided a detailed description.
- (3) We added Fig. R2-7 into Supplementary Figure 3 and provided a detailed description.

9. The influence of the oxide layer on the joined surface is not fully explained. The details on the decomposition behavior of oxide phases such as ZrO_2 etc. should be presented.

Response:

Thank you for your professional concern. In response to these issues, we conducted a more detailed TEM characterization (**Fig. R2-8**). The results revealed that the actual mechanism involved the fragmentation of the oxide layer into numerous nano-sized particles and dispersed within the MGs matrix. Through EDS analysis conducted with TEM, we also identified the oxide particles of the two components in the heterogeneous joined sample (**Fig. R2-8b, c**).

The dispersion of the oxide layer was caused by the special effect of the fluid-like MGs under ultrasonic vibration. When high-intensity ultrasonic vibration acts on liquid, cavitation and acoustic streaming will be generated^{7,8}, which produces high-intensity shock waves, strong convection, and turbulence in the fluid. It is well-known that metal oxide layers are much more brittle¹⁴, so it is easy to be crushed under the above action. As a result, the oxide layer was broken into small particles and gradually dispersed in the MGs matrix. After the oxide layer dispersed, the diffusion barrier disappears, and the fresh flow surface can come into contact and mix vigorously under ultrasonic vibration (**Fig. R2-9**). We have added the relevant discussion in the revised manuscript.

Fig. R2-8. a, The high angle annular dark field (HAADF) images at the interface of heterogeneous joined sample, which show dispersed oxide particles and mixed interfaces. **b**, Elemental analysis of $Zr_{55}Cu_{30}Al_{10}Ni_5$ MG oxide particles. **c**, Elemental analysis of $TiZrHfBeNi$ MG oxide particles.

Fig. R2-9. Schematic diagram illustrating the mechanism of oxide layer fragmentation and interfacial bonding.

Changes made in the revised manuscript:

- (1) Page 12, Line 233

We have **added** paragraphs to describe the observation of the oxide layer: “More importantly, large number of dispersed oxide particles were observed around the interface, proving the breakdown of the oxide layer. Except a few large particles, most of these particles have been crushed to considerably smaller size, more analysis of oxide particles as shown in the supplementary Fig.4”.

- (2) We have **added** Fig R2-8 as Supplementary Figure 4 and provided a detailed description.

- (3) Page 15, Line 317

We added Fig. R2-9 into Fig. 4 and provided a detailed description: “These effects promote the broken and dispersion of the brittle oxide layer. On the other hand, the ultrasonic-induced flow provides strong mixing of two fresh high-energy state fluid MG”.

10. The uniform distribution mechanism of oxygen at the low temperature into the

matrix part should be presented. The diffusivity of oxygen to the matrix phase at low temperatures is very low and hence the homogeneous distribution of oxygen from ZrO_2 oxide seems to be difficult.

Response:

Thank you for your professional concern. Based on detailed TEM observations, we found that the oxide layer did not uniform distribution after diffusion. In fact, in this work, due to the effects of cavitation and acoustic streaming in softened MG induced by ultrasonic vibration, the oxide layer was completely broken into small particles and dispersed throughout the matrix. The related discussion can also be found in the above **Response 9** for details.

11. How much is the oxygen content at the interface between the ZrCuAlNi BMG and the high entropy BMG?

Response:

Thank you for your concern. In the interface between the ZrCuAlNi BMG and the high entropy BMG, EDS mapping and line scan showed a uniform distribution of oxygen elements with a content of about 1.5% (**Fig R2-10**).

Fig. R2-10. Element analysis of heterogeneous joined interface under SEM.

Changes made in the revised manuscript:

(1) We have **added** Fig R2-10 as Supplementary Figure 3 and provided a detailed description.

12. The more detailed information on the joined interface structure should be presented. The authors have claimed that the joining occurs through atomic diffusion. However, there are no information on the diffusion distance and on the possibility of the formation of a new BMG phase region with different composition. Even in the high-resolution TEM image, there are no data on new image contrast region. The authors should present the change in alloy composition distribution at the joined interface by nanobeam EDX analytical method. Some previous papers have presented such finely analyzed data at the joined interface part.

Response:

Thank you for your professional and valuable concern.

To investigate the joined interface structure, a more detailed TEM characterization was conducted. We can find nanoscale mixing zone was observed between the two MGs (**Fig. R2-11a**). A selected interface was observed in **Fig. R2-11b**, where the mutual diffusion of elements at the two-phase interface can be clearly seen, and the diffusion layer can reach up to about 50 nm. Meanwhile, nanobeam EDX with scanning diameter of 2 nm was used to analysis multiple points at the interface (**Fig. R2-11b, c**), in which the quantitative element analysis proved the existence of the diffusion layer. We have added the relevant content in the revised manuscript.

Fig. R2-11. a, b. The high angle annular dark field (HAADF) images at the interface of heterogeneous join. **c.** The nano-beam EDX through the diffusion layer. The scanning point is shown in b.

Changes made in the revised manuscript:

(1) We added Fig. R2-11 into Fig. 3 and provided a detailed description.

(2) Page 12, Line 236:

We have **added** the paragraphs to research the joined interface: “A selected interface was observed in Fig. 3i, where the mutual diffusion of elements at the two-phase interface can be clearly seen, and the diffusion layer can reach up to about 50 nm. Meanwhile, nanobeam EDX with scanning diameter of 2 nm was used to analysis multiple points at the interface (Fig. 3j), in which the quantitative element analysis at the interface proved the existence of the diffusion layer.”

(3) Page 23, Line 492:

We have **added** the sentences “The composition distribution at the joined interface was conducted using nano-beam EDX with a beam diameter of 2 nm” in Methods

13. The tensile test was made for the thin sheet with a thickness of 0.5 mm. Is it possible to measure the tensile testing data for the thicker sheet sample?

Response:

Thank you for your professional concern. We prepared 2 mm-thick ZrCuAlNi bulk metallic glasses (BMG) using the thickest mold available to us, and retested its tensile properties after joining. The thickness of the tensile parts prepared after grinding is 1.5 mm. The results showed that the tensile strength of the 2 mm-thick sample was 1112

MPa, which was 70% of that of the original sample (**Fig. R2-12**). As for the thicker sample, it is more likely to bring in defect during the joining process (**Fig. R2-12**). The reduction of the tensile strength could be caused by the joining defects in the thicker sample.

Fig. R2-12. Stress-strain curves and corresponding break surface morphology under room temperature tensile.

14. There is no explanation on the mutual relation between the tensile fracture surface (site) and the joined interface part. The clarification on this relation is extremely important for the engineering usefulness of this welding process.

Response:

Thank you for your valuable comments. In fact, we find that the tensile fracture site is still near the joined interface, and the traces of joined interface can be found in several samples.

Different energies applied during ultrasonic joining will result in different fracture surface and thus affect the tensile strength. To confirm the relationship between the tensile fracture surface (site) and the joined interface part, we observed different joined

sample (**Fig. R2-13**). On the fracture surface of the 80 J sample (519 MPa), large number of interfaces with poor joining quality can be observed. However, the fracture surface of the highest strength (150 J, 1522 MPa) sample could only be found with a few poor joined interfaces, which demonstrated the fracture surface still remained near the interface. In addition, we think that if there are too many poor joined interfaces exist, it will lead to a worse strength. According to the suggestion, we have added related content in the revised manuscript.

Fig. R2-13. Break surface morphology of Zr-based jointed sample. Due to the use of butt joint (sheet) samples in the tensile tests, the overall value of welding energy applied is relatively small.

Changes made in the revised manuscript:

(1) We have **added** Fig R2-13 as Supplementary Figure 5 and provided a detailed description.

(2) Page 17, Line 348:

We have **added** the paragraphs to describe the fractured surface: “Actually, small amounts of poor joined interfaces could be found in break surface morphology, as shown in Supplementary Fig. 5, which demonstrated the fracture surface still

remained near the interface. If the welding energy is reduced, the tensile strength will decrease and the poor joined interfaces will become more pronounced.”

15. According to some previous papers on the fractured surface for the joined BMGs, the joined interface in the two BMGs has been observed clearly in the fracture surface SEM images, being completely different from the present fracture surface image. The reason for the inconsistent data on the fracture surface should be presented.

Response:

Thank you for your valuable comments. We observed the fracture morphology of joined samples in the literature¹⁵ (**Fig R2-14**), in which a large number of interfaces found. In fact, this is similar to the fracture surface in our work, but our work has fewer interfaces (**Fig. R2-13**). If there are too many poor joined interfaces exist, the sample may exhibit a worse strength. In addition, poor joined interfaces are difficult to disappear completely, so the fracture will occur near the interface.

Chen W., *et al.* (2014).

Fig. R2-14. Break surface morphology of previous papers

- 15 Chen, W., Liu, Z. & Schroers, J. Joining of bulk metallic glasses in air. *Acta Materialia* **62**, 49-57, (2014).

Changes made in the revised manuscript:

- (1) Page 17, Line 348:

We have **added** the paragraphs to describe the fractured surface: “Actually, small

amounts of poor joined interfaces could be found in break surface morphology, as shown in Supplementary Fig. 5, which demonstrated the fracture surface still remained near the interface.”

16. The details for the specimen part used for the measurement of corrosion behavior should be presented.

Response:

Thank you for your comments. Our corrosion samples are made of 5 mm butt-joined cylindrical specimens. The 5 mm cylindrical joined specimens were cut in half, and their longitudinal-section was selected as the surfaces for corrosion testing, as shown in **Fig.R2-15**. Meanwhile, all test surfaces measuring 5×6 mm in size. Before the corrosion test, the specimens were mechanically polished to a mirror finish, then degreased in acetone, washed in distilled water, dried in air, and further exposed to air for 24 hours to obtain good reproducibility.

Fig. R2-15. The joined specimen used for the measurement of corrosion behavior.

Changes made in the revised manuscript:

(1) Page 25, Line 517:

We have **modified** the details for the corrosion test specimen: “The cross-sectional surface of the butt-joint MGs ($Zr_{55}Cu_{30}Al_{10}Ni_5$) of 5 mm cylindrical was used as the corrosion surface” into “The 5 mm cylindrical joined specimens were cut in half, and their longitudinal-section was selected as the surfaces for corrosion testing, as shown in Supplementary Figure 12.”.

(2) We have **added** Fig R2-15 as Supplementary Figure 12 and provided a detailed description.

17. The difference in the corrosion resistance seems to be recognized among the three parts, namely, ZrCuAlNi BMG, joined part and high entropy BMG. The difference in the corrosion resistance among the three phase regions should be presented.

Response:

Thank you for your comments. Here we compare the differences in corrosion resistance among the three-phase regions by observing the morphology of heterogeneous joined sample at different corrosion stages. In the early stages of corrosion, pitting occurred extensively on one side of the ZrCuAlNi phase, but there was no significant pitting on the ZrCuAlNi phase near the interface (**Fig R2-16 a-c**), which is similar to the results of homogeneous joining in Fig. 5h-k. In the middle stages of corrosion, after one side of the ZrCuAlNi phase was fully corroded, we found that the two-phase mixed zone still contained remaining ZrCuAlNi phase (**Fig R2-16 d-f**), demonstrating that the mixed zone is more corrosion-resistant. In the later stages of corrosion, after the high entropy BMG also suffered severe corrosion, it is worth noting that the high entropy BMG near the interface was not corroded (**Fig R2-16 e, g**). Meanwhile, the ZrCuAlNi BMG in the mixed region was completely corroded, leaving only the high entropy BMG phase (**Fig R2-16 h**).

In summary, the portions closer to the interface will exhibit better corrosion resistance, particularly in the mixing zone. This is attributed to the MG relaxation caused by the absorption of more ultrasonic vibration energy at the interface We have added related content in the revised manuscript.

Fig. R2-16. The corrosion morphology observation of heterogeneous joined samples. **a-c,** The SEM images of the early stages in corrosion testing. **d-f,** The SEM images of the middle stages in corrosion testing. **e-h,** The SEM images of the later stages in corrosion testing.

Changes made in the revised manuscript:

(1) Page 18, Line 370:

We have **added** the details for the corrosion situation in heterogeneous joined samples: “At the same time, the corrosion performance of heterogeneous joined samples was also analyzed, as shown in Supplementary Fig. 6. Although TiZrHfBeNi has much better corrosion resistance than $Zr_{55}Cu_{30}Al_{10}Ni_5$, we also found that the portions closer to the interface will exhibit better corrosion resistance, particularly in the two-phase mixing zone.”

(3) We have **added** Fig R2-16 as Supplementary Figure 6 and provided a detailed description.

18. According to many previous data on the corrosion resistance, the corrosion resistance of BMGs decreases upon annealing. The present data appears to be opposite. The inconsistent reason should be explained.

Response:

Thank you for your comments. In fact, we find extensive literature has demonstrated that annealing leads to improved corrosion performance of MGs, benefiting from a more stable amorphous structure after annealing. Here, we list some typical papers¹⁶⁻¹⁹ and the corrosion performance graphs (Fig R2-17).

Fig. R2-17. Examples of corrosion resistance improvement by annealing.

16 Jindal, R. *et al.* Effect of annealing below the crystallization temperature on the corrosion behavior of Al–Ni–Y metallic glasses. *Corrosion science* **84**, 54-65 (2014).

17 Zhou, M., Hagos, K., Huang, H., Yang, M. & Ma, L. J. J. o. N.-C. S. Improved mechanical properties and pitting corrosion resistance of Zr₆₅Cu_{17.5}Fe₁₀Al_{7.5} bulk metallic glass by isothermal annealing. *Journal of Non-Crystalline Solids* **452**, 50-56 (2016).

18 Huang, C. *et al.* Improvement of bio-corrosion resistance for Ti₄₂Zr₄₀Si₁₅Ta₃ metallic glasses in simulated body fluid by annealing within supercooled liquid region. *Materials Science and Engineering: C* **52**, 144-150 (2015).

19 Liang, D. *et al.* Investigation of the structural heterogeneity and corrosion performance of the annealed Fe-based metallic glasses. *Materials* **14**, 929 (2021).

19. The similar data have already been presented in Scripta Mater. in 2020. The authors

should clarify the difference between the previous data and the present data. This manuscript seems to be a kind of technical report. The more detailed interpretation and discussion should be required to improve the quality of this manuscript.

Response:

Thank you for your valuable and kind comments. This work actually reports a series of different and new application scenarios for MGs. First, the work of Scripta Mater. in 2020 reported on the joining of MG in air, while other joining methods can also be used to weld MGs in air. Impressively, the present work realized joining in liquid environments, in which almost all of other joining methods cannot be utilized. Thus, this work presents a promising method to realize joining under challenging environments, such as offshore, polar, oil-gas fields and space. Secondly, this work presents the successful joining of similar and even dissimilar MGs including high-strength Zr-based MG that is important for commercial applications. In comparison, only La-based MG were joined in Scripta Mater., which is not a suitable alloy system for practical use.

To improve the quality of this manuscript, we have clarified the mechanism of ultrasonic-induced softening and welding, provided the accurate temperature measurement, as well as conducted the joining interface analysis, etc. We sincerely hope the revisions have addressed the concerns of the reviewer.

Response to Reviewer #3

This paper reports an ultrasonic vibration based welding of metallic glass in various liquids. What they showed in the welding demos as well as the comparable strength of welded sample with the base alloy are very impressive. The experiments are in general vigorously done and the characterization are sound. It may be appropriate to publish in Nature Communications. I have several questions, which are described in the following:

Response:

Thank you for your positive comments.

1. It seems that the ultrasonic vibration welding is very suitable for liquid environment welding, it is suggested to add discussion on the challenge of ultrasonic vibration based underwater welding by comparison with the traditional ultrasonic vibration welding in air.

Response:

Thank you for your professional and valuable concern. We have added discussion on the challenge of ultrasonic vibration based underwater welding by comparison with the traditional ultrasonic vibration welding in air.

Changes made in the revised manuscript:

(1) Page 5, Line 89:

We have **added** the paragraphs to discuss the challenge of ultrasonic vibration: “In recent years, a novel joining method has emerged - ultrasonic vibration joining (UVJ) technology^{38,39}. Generally, ultrasonic-related technology was used in the field of welding for plastics^{40,41} and low melting metals⁴¹, except for these, more often as an auxiliary technology to other welding methods⁴²⁻⁴⁴, which is also difficult to be applied in liquid environments due to the temperature-dependent mechanism. However, due to the characteristics of low temperature requirement, fast joining process, and simple operation when joining MGs³⁹, UVJ technology has great potential for realize joining MGs in liquid environments.”

The references include:

38 Ma, J. *et al.* Fast surface dynamics enabled cold joining of metallic glasses. *Science advances* **5**, eaax7256 (2019).

39 Li, X., *et al.* Cold joining to fabricate large size metallic glasses by the ultrasonic vibrations. *Scripta Materialia* **185**, 100-104 (2020).

40 Benatar, A. & Marcus, M. *Power Ultrasonics (Second Edition)* (Woodhead Publishing, Cambridge, 2023).

41 Tsujino, J. *et al.* New methods of ultrasonic welding of metal and plastic materials. *Ultrasonics* **34**, 177-185 (1996).

42 Cui, Y., Xu, C. & Han, Q. Effect of ultrasonic vibration on unmixed zone formation. *Scripta materialia* **55**, 975-978 (2006).

43 Rahmi, M. & Abbasi, M. Friction stir vibration welding process: modified version of friction stir welding process. *The International Journal of Advanced Manufacturing Technology* **90**, 141-151 (2017).

44 Kumar, S., Wu, C., Padhy, G. & Ding, W. Application of ultrasonic vibrations in welding and metal processing: A status review. *Journal of manufacturing processes* **26**, 295-322 (2017).

2. Is there any difference in the welding mechanism by using ultrasonic vibration welding in air and in liquid? It is suggested to compare the welding performance of joined samples in air and in different liquids (pure water, seawater, alcohol and liquid nitrogen), but with the same ultrasonic vibration welding conditions.

Response:

Thank you for your valuable comments. In the present work, the mechanism of welding in air and under-liquids is presented similarly: Firstly, the fluidlike behavior of MGs near the interface is triggered, which is caused by the dynamic heterogeneity and expansion of liquid-like regions of MG. Subsequently, ultrasonic vibration generates cavitation and acoustic streaming in flowing MG, dispersing the oxide layer and promoting strong mixing between the two types of MG.

The fluidlike behavior of MG induced by ultrasonic vibration is the basis mechanism for both joining in air or liquid environments, and it is not dependent on temperature rise. Although the underlying mechanism bears similarity, notable distinctions exist that render this study a promising approach for under-liquid joining:

1. Various liquid environments allow for more applications;
2. Due to part energy dissipation in the liquid, joining in liquid environments requires more energy than in air, as shown in **Fig.R3-1**.

Fig. R3-1. The joining performance of heterogeneous joined samples in air and in different liquids (pure water, seawater, alcohol and liquid nitrogen). The inset shows the line scan of the oxygen element at the interface.

Changes made in the revised manuscript:

- (2) We have **added** Fig R3-1 as Supplementary Figure 3 and provided a detailed description.

(3) Page 9, Line 175:

We have **added** the paragraphs to describe the welding performance in air and in different liquids: “Meanwhile, the joining in liquid environment was compared with air, as shown in the supplementary materials. Due to the dissipation of ultrasonic energy, joining in liquid environments requires more energy than in air. The results showed that under appropriate welding parameters, all environments (include air) can ultimately achieve completely defect-free joining.”

3. The Referee is very interested in the welding mechanism, the authors attributed it to the breakage of oxide layer and atom diffusion. It is hard to imagine that the oxide layer can break into atomic scale size (700 J in Fig. 3b, referring to such as: Acta Mater., 62, 49-57, 2014), is it possible that just atoms diffusion into oxide layer and significantly dilute the oxygen content the oxide layer?

Response:

Thank you for your professional concern. In response to these issues, we conducted a more detailed TEM characterization (**Fig. R3-2**). The results revealed that the actual mechanism is the oxide layer break into numerous nano-sized particles and dispersed within the MGs matrix. Through EDS analysis conducted with TEM, we also identified the oxide particles of the two components in the heterogeneous joining (**Fig. R3-2b, c**).

The dispersion of the oxide layer was caused by the special effect of the fluid-like MG under ultrasonic vibration. It is well-known that metal oxide layers are much more brittle¹⁴. When high-intensity ultrasonic vibration acts on liquid, cavitation and acoustic streaming will be generated^{7,8}, which produces high-intensity shock waves, strong convection, and turbulence in the fluid (**Fig. R3-3**). As a result, the oxide layer was broken into small particles and gradually dispersed in the MGs matrix. We have added the relevant discussion in the revised manuscript.

Fig. R3-2. **a**, The high angle annular dark field (HAADF) images at the interface of heterogeneous joined sample, which show dispersed oxide particles and mixed interfaces. **b**, Elemental analysis of $Zr_{55}Cu_{30}Al_{10}Ni_5$ MG oxide particles. **c**, Elemental analysis of TiZrHfBeNi MG oxide particles.

Fig. R3-3. Schematic diagram illustrating the mechanism of oxide layer fragmentation and interfacial bonding.

- 7 Eskin, G. (Moscow, Russia-Gordon and Breach science publisher, 1998).
- 8 Cui, Y., Xu, C. & Han, Q. Effect of ultrasonic vibration on unmixed zone formation. *Scripta materialia* 55, 975-978 (2006).
- 14 Picqué, B., Bouchard, P.-O., Montmitonnet, P. & Picard, M. Mechanical behaviour of iron oxide scale: experimental and numerical study. *Wear* 260, 231-242 (2006).

Changes made in the revised manuscript:

- (1) Page 12, Line 233

We have **added** paragraphs to describe the observation of the oxide layer: “More importantly, large number of dispersed oxide particles were observed around the interface, proving the breakdown of the oxide layer. Except a few large particles, most of these particles have been crushed to considerably smaller size, more analysis of oxide particles as shown in the supplementary Fig.4”.

- (2) We have **added** Fig R3-2 as Supplementary Figure 4 and provided a detailed description.

- (3) Page 15, Line 317

We **added** Fig. R3-3 into Fig. 4 and provided a detailed description: “These effects promote the broken and dispersion of the brittle oxide layer. On the other hand, the ultrasonic-induced flow provides strong mixing of two fresh high-energy state fluid MGs, which promotes full contact of the interface and generates an irregular interface.”.

- (4) Page 5, Line 86

We have **added** References: *Acta Mater.*, 62, 49-57, 2014.

4. The authors attribute the good welding performance to the fast surface dynamics induced fluidlike behavior, what’s the specific surface dynamic behavior (liquid or BMG related?) and underlying mechanism? Is it the unique behavior of ultrasonic vibration welding in liquid or not? More evidence and details are needed.

Response:

Thank you for your valuable suggestion. “The fast surface dynamics induced fluidlike behavior” is the unique property of MGs, which means the atomic mobility is much faster on the surface of MGs compared with the inner position. The depth of such liquid-like layer could be several atomic layers, much thicker than the crystalline alloys. After a more insightful investigation, we think “the dynamic heterogeneity of MGs and the expansion of the liquid-like region induced fluidlike behavior” could be more accurate. The detailed mechanism is described as follows:

The MGs consist of liquid-like and solid-like regions, a phenomenon known as structural heterogeneity or dynamic heterogeneity¹. The solid-like region forms elastic network, while the liquid-like region enveloped by the network acts as a viscous flow unit that dissipates energy. In BMGs, local shear events induced by external forces have been shown to cause expansion of the liquid-like region after absorbing stress, which promotes the growth and formation of the liquid-like region^{20,21}. Under cyclic loading, due to the absence of sufficient time for stress relaxation, the pressure accumulates, causing expansion of the liquid-like regions and then present fluidlike behavior^{2,3} (**Fig R3-4a**). The dynamic scanning probe microscope (DSPM) image of the $Zr_{55}Cu_{30}Al_{10}Ni_5$ MG as shown in **Fig R3-4b-d**, therein the viscoelastic loss tangent ($\tan\delta$) is proven to correspond to internal friction. When the frequency increase to the ultrasonic vibrations, the $\tan\delta$ value increases rapidly and all regions exhibit higher than 0.22, indicating the activation and extensive expansion of the liquid-like regions. We have added the relevant discussion in the revised manuscript. We have added the relevant discussion in the revised manuscript.

Due to the softening mechanism is not based on temperature, the fluidlike behavior of MGs under ultrasonic vibration is independent of the environment and is closely related to the dynamic non-uniformity of BMGs and cyclic stress under ultrasonic vibration. Evidence indicates that the same MG can be softened in both air and liquid environments (**Movies R1-2**).

Fig. R3-4. Softening mechanism of MG.

- 1 Qiao, J. *et al.* Structural heterogeneities and mechanical behavior of amorphous alloys. *Progress in Materials Science* **104**, 250-329 (2019).
- 2 Galindo-Torres, S., Zhang, X. & Krabbenhoft, K. Micromechanics of liquefaction in granular materials. *Physical Review Applied* **10**, 064017 (2018).
- 3 Li, X. *et al.* Ultrasonic plasticity of metallic glass near room temperature. *Applied Materials Today* **21**, 100866 (2020).
- 20 Wang, Y.-J., Jiang, M., Tian, Z. & Dai, L. Direct atomic-scale evidence for shear–dilatation correlation in metallic glasses. *Scripta Materialia* **112**, 37-41 (2016).
- 21 Peng, C. *et al.* Deformation behavior of designed dual-phase CuZr metallic glasses. *Materials & Design* **168**, 107662 (2019).

Changes made in the revised manuscript:

- (1) Page 13, Line 272
We have **added** ultrasonic-induced softening mechanism in Results and discussion.
- (2) We have **added** Fig R3-4 in Fig.4.
- (3) Page 2, Line 27

We have **revised** “the fast surface dynamics induced fluid-like behavior” as “The dynamic heterogeneity and liquid-like region expansion induced fluid-like behavior”

5. Fig. 3 clearly shows the effect of oxide layer on the welding performance. I am wondering that is ultrasonic welding performance better in liquid environment with less oxygen content than in air? If so, is the required threshold input energy (for example, 700 J in Fig. 3a-b) also lower in liquid than in air?

Response:

Thank you for your valuable and kind comments. We conducted a comparison of different energy in various environments. The results showed that under appropriate welding parameters, all environments can ultimately achieve completely defect-free joining. However, due to the dissipation of part energy in liquid, joining in liquid environments requires more energy than that in air (**Fig.R3-5**). Meanwhile, the oxygen content at the interface is very similar in all environments.

Fig. R3-5. The joining performance of joined samples in air and in different liquids (pure water, seawater, alcohol and liquid nitrogen). The inset shows the line scan of the oxygen element at the interface.

Changes made in the revised manuscript:

- (1) We have **added** Fig R3-5 as Supplementary Figure 3 and provided a detailed description.

6. The authors attributed the honeycomb-like structure in the non-interface area (Fig. 4j) to intense corrosion, but it seems more like the vein-pattern of typical fractured BMG surface to the Referee (referring to such as: *Acta Mater.*, 62, 49-57, 2014), I think this is important to the mechanism understanding and suggest the authors to clarify it.

Response:

Thank you for your comments. We also think the 'honeycomb'-like structure is similar to the fractured BMG surface, but it is recognized as a typical pitting morphology. Here, similar morphologies from other literature²²⁻²⁴ are cited to demonstrate that this pitting morphology is typical, as shown in **Fig R3-6**. We re-photographed the pitting pattern to make the data clearer, as shown in **Fig.R3-7**.

Fig. R3-6. Examples of 'honeycomb'-like pitting morphology in other literature.

Fig. R3-7. Clearer pitting morphology in this work.

- 22 LIU L, QIU C, ZOU H, et al. The effect of the microalloying of Hf on the corrosion behavior of ZrCuNiAl bulk metallic glass [J]. *Journal of Alloys and Compounds*, 2005, **399**(1-2): 144-8.
- 23 ZHANG Y, YAN L, ZHAO X, et al. Enhanced chloride ion corrosion resistance of Zr-based bulk metallic glasses with cobalt substitution [J]. *Journal of Non-Crystalline Solids*, 2018, **496**: 18-23.
- 24 HUA N, LIAO Z, WANG Q, et al. Effects of crystallization on mechanical behavior and corrosion performance of a ductile $Zr_{68}Al_8Ni_8Cu_{16}$ bulk metallic glass [J]. *Journal of Non-Crystalline Solids*, 2020, **529**: 119782.

Changes made in the revised manuscript:

- (1) We have **modified** the Fig. R3-7 and related information as. Fig. 5j.

REVIEWERS' COMMENTS

Reviewer #1 (Remarks to the Author):

Thanks for the detailed revision and additional data as well as realized experiments.

Reviewer #3 (Remarks to the Author):

Since all reviewers' comments have been well addressed, and the manuscript is greatly improved, I suggest to publish this interesting paper after minor revision.

1. Line 232 at page 11, "...expansion of the liquid-like region after absorbing stress", it is rarely referred to as "absorbing stress", you can say "absorbing energy" since the expansion of the liquid-like region originates from plastic deformation.
2. The authors show nice TEM results on the nano-sized oxide particles (Fig. R3-2b,c), it is strange that there seems to be no enrichment of metal elements in the oxygen rich regions, then what kind of oxides forms?
3. There are still spelling, grammar, and typos in the manuscript. For example, line 121 at page 6, the number and trigonometric function should use regular bodies. Similar issue at page 13, line 259.

Responses to Reviewers' Reports on NCOMMS-22-50999B

We would like to thank the referees for reviewing our manuscript entitled “Joining of metallic glasses in liquid via ultrasonic vibrations” (NCOMMS-22-50999B). We are grateful that all reviewers raised positive comments on this work, and provided constructive suggestions to help us to substantially improve the quality of this work.

In the following, we provide below a point-to-point reply to all comments of the referees. All the concerns have been properly addressed, we also included the comments below followed by our responses in blue for your convenience.

Response to Reviewer #1

Thanks for the detailed revision and additional data as well as realized experiments.

Response:

We thank the reviewers for the positive comments.

Response to Reviewer #3

Since all reviewers' comments have been well addressed, and the manuscript is greatly improved, I suggest to publish this interesting paper after minor revision.

Response:

Thank you for your positive comments.

1. Line 232 at page 11, “...expansion of the liquid-like region after absorbing stress”, it is rarely referred to as “absorbing stress”, you can say “absorbing energy” since the expansion of the liquid-like region originates from plastic deformation.

Response:

Thank you for your professional and valuable concern. Based on your

suggestions, we have made the modifications to the relevant description.

Changes made in the revised manuscript:

(1) Page 12, Line 234

We have **modified** the verb 'stress' to 'energy'.

2. The authors show nice TEM results on the nano-sized oxide particles (Fig. R3-2b,c), it is strange that there seems to be no enrichment of metal elements in the oxygen rich regions, then what kind of oxides forms?

Response:

Thank you for your professional concern. In fact, due to the enrichment of oxygen content, the metallic elements in oxide particles are inevitably more depleted compared to the non-oxidized region, as shown in the EDS mapping scanning results (**Fig. R1**).

To check the composition of those oxide particles, we have conducted more accurate nanobeam EDX spectrum analysis. The results indicate that all elements belong to the constituent elements of raw material. One type of oxide particles is primarily composed of Zr, Cu, Ni, Al (**Fig. R2a**), while another oxide particles is primarily composed of Zr, Ti, Hf, Ni, Cu (**Fig. R2b**).

Fig. R1 a The EDS mapping of $Zr_{55}Cu_{30}Al_{10}Ni_5$ MG oxide particles. **b** The EDS mapping of $TiZrHfBeNi$ MG oxide particles.

Fig. R2 a, b. The EDX spectrum of two types of oxide particles. The red box represents the scanning area.

3. There are still spelling, grammar, and typos in the manuscript. For example, line 121 at page 6, the number and trigonometric function should use regular bodies. Similar issue at page 13, line 259.

Response:

Thank you for your valuable suggestions. According your suggestions, we have made the modifications in the relevant description.

Changes made in the revised manuscript:

(1) Page 6, Line 121

We have **modified** the number and trigonometric function ' $d(t) = A\sin(2\pi f \cdot t)$ ' to ' $d(t) = A\sin(2\pi f \cdot t)$ '.

(2) page 13, line 263

We have **modified** the number and trigonometric function ' $\tau = 1/(\omega \tan \delta)$ ' to ' $\tau = 1/(\omega \tan \delta)$ '.